# DEPTH ANYTHING 3:
# RECOVERING THE VISUAL SPACE FROM ANY VIEWS

**Haotong Lin**[*1]   **Sili Chen**[*1]   **Jun Hao Liew**[*1]   **Donny Y. Chen**[*1]   **Zhenyu Li**[1]   **Yang Zhao**[1]
**Sida Peng**[2]   **Hengkai Guo**[3]   **Xiaowei Zhou**[2]   **Guang Shi**[1]   **Jiashi Feng**[1]   **Bingyi Kang**[*1†]
[1]ByteDance Seed   [2]Zhejiang University   [3]ByteDance Intelligent Creation
[*]Equal contribution   [†]Project lead & Corresponding author

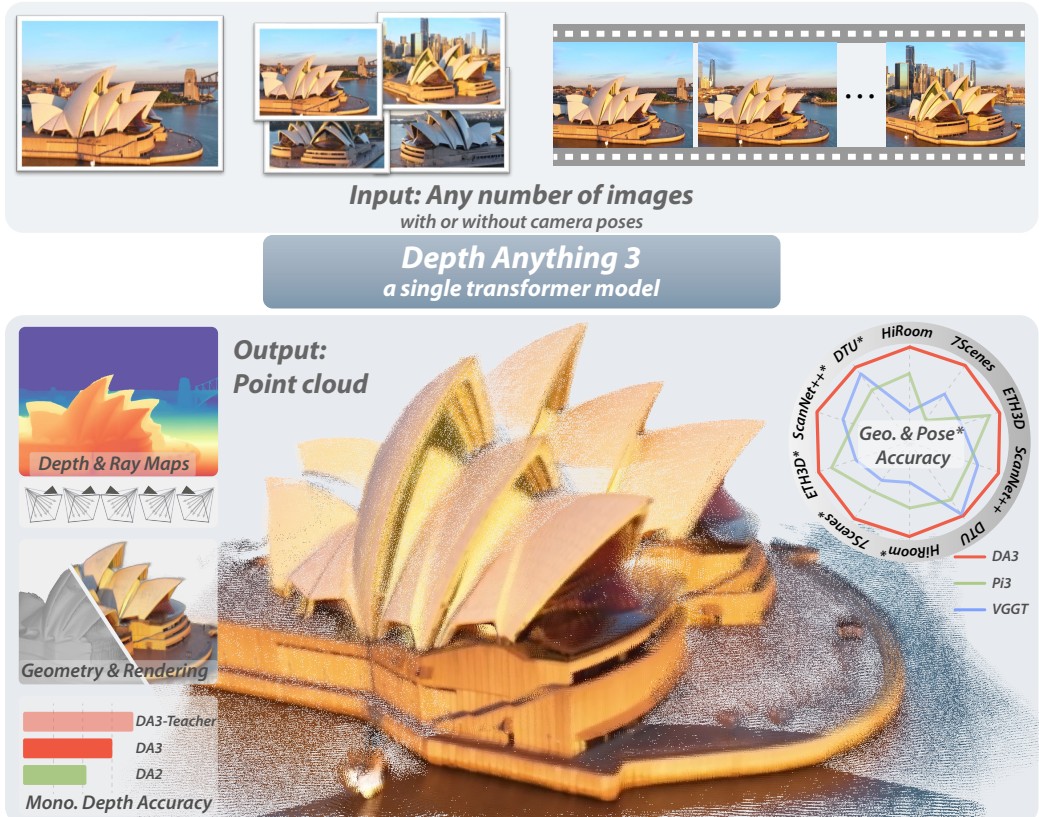

Figure 1: Given any number of images and optional camera poses, **Depth Anything 3** reconstructs the visual space, producing accurate depth and ray maps that fuse into a consistent point cloud. It substantially outperforms VGGT in multi-view geometry and pose accuracy; with monocular inputs, it also surpasses Depth Anything 2 while matching its detail and robustness.

## ABSTRACT

We present Depth Anything 3 (DA3), a model that predicts spatially consistent geometry from an arbitrary number of visual inputs, with or without known camera poses. In pursuit of minimal modeling, DA3 yields two key insights: a single plain transformer (*e.g.*, vanilla DINO encoder) is sufficient as a backbone without architectural specialization, and a singular depth-ray prediction target obviates the need for complex multi-task learning. Through our teacher-student training paradigm, the model achieves a level of detail and generalization on par with Depth Anything 2 (*DA2*). We establish a new visual geometry benchmark covering camera pose estimation, any-view geometry and visual rendering. On this benchmark, DA3 sets a new state-of-the-art across all tasks, surpassing prior SOTA *VGGT* by an average of 35.7% in camera pose accuracy and 23.6% in geometric accuracy. Moreover, it outperforms *DA2* in monocular depth estimation. All models are trained exclusively on public academic datasets.

# 1 INTRODUCTION

The ability to perceive and understand 3D spatial information from visual input is a cornerstone of human spatial intelligence (Arterberry and Yonas, 2000) and a critical requirement for applications like robotics and mixed reality. This fundamental capability has inspired a wide array of 3D vision tasks, including monocular depth estimation, Structure from Motion (Snavely et al., 2006), Multi-View Stereo (Seitz et al., 2006) and Simultaneous Localization and Mapping (Mur-Artal et al., 2015). Despite the strong conceptual overlap between these tasks—often differing by only a single factor, such as the number of input views—the prevailing paradigm has been to develop highly specialized models for each one. While recent efforts (Wang et al., 2024c; 2025a) have explored unified models to address multiple tasks simultaneously, they typically suffer from several key limitations: they often rely on complex, bespoke architectures, are trained via joint optimization over tasks from scratch, and consequently cannot effectively leverage large-scale pretrained models.

In this work, we step back from established 3D task definitions and return to a more fundamental goal inspired by human spatial intelligence: recovering 3D structure from arbitrary visual inputs, be it a single image, multiple views of a scene, or a video stream. Forsaking intricate architectural engineering, we pursue a minimal modeling strategy guided by two central questions. First, *is there a minimal set of prediction targets, or is joint modeling across numerous 3D tasks necessary?* Second, *can a single plain transformer suffice for this objective*? Our work provides an affirmative answer to both. We present Depth Anything 3, a single transformer model trained exclusively for joint **any-view depth and pose estimation** via a specially chosen ray representation. We demonstrate that this minimal approach is sufficient to reconstruct the visual space from any number of images, with or without known camera poses.

Depth Anything 3 formulates the above geometric reconstruction target as a dense prediction task. For a given set of $N$ input images, the model is trained to output $N$ corresponding depth maps and ray maps, each pixel-aligned with its respective input. The architecture to achieve this begins with a standard pretrained vision transformer (e.g., Oquab et al. 2023), as its backbone, leveraging its powerful feature extraction capabilities. To handle arbitrary view counts, we introduce a key modification: an input-adaptive cross-view self-attention mechanism. This module dynamically rearranges tokens during the forward pass in selected layers, enabling efficient information exchange across all views. For the final prediction, we propose a new dual DPT head designed to jointly outputs both depth and ray values, by processing the same set of features with distinct fusion parameters. To enhance flexibility, the model can optionally incorporate known camera poses via a simple camera encoder, allowing it to adapt to various practical settings. This overall design results in a clean and scalable architecture that directly inherits the scaling properties of its pretrained backbone.

We train Depth Anything 3 via a teacher-student paradigm to unify diverse training data, which is necessary for a generalist model. Our data sources include varied formats like real-world depth camera captures (e.g., Baruch et al. 2021), 3D reconstruction (e.g., Reizenstein et al. 2021), and synthetic data, where real-world depth may be of poor quality (Fig. 7). To resolve this, we adopt a pseudo-labeling strategy inspired by prior works (Yang et al., 2024a;b). Specifically, we train a powerful teacher monocular depth model on synthetic data to generate dense, high-quality pseudo-depth for all real-world data. Crucially, to preserve geometric integrity, we align these dense pseudo-depth maps with the original sparse or noisy depth. This approach proved remarkably effective, significantly enhancing label detail and completeness without sacrificing the geometric accuracy.

To better evaluate our model and track progress in the field, we establish a comprehensive benchmark for assessing geometry and pose accuracy. The benchmark comprises 5 distinct datasets, totaling over 89 scenes, ranging from object-level to indoor and outdoor environments. By directly evaluating pose accuracy across scenes and fusing the predicted pose and depth into a 3D point cloud for accuracy assessment, the benchmark faithfully measures the pose and depth accuracy of visual geometry estimators. Experiments show that our model achieves state-of-the-art performance on 18 out of 20 settings. Moreover, on standard monocular benchmarks, our model outperforms Depth Anything 2 (Yang et al., 2024b).

To further demonstrate the fundamental capability of Depth Anything 3 in advancing other 3D vision tasks, we introduce a challenging benchmark for feed-forward novel view synthesis (FF-NVS), comprising over 160 scenes. We adhere to the minimal modeling strategy and fine-tune our model with an additional DPT head to predict pixel-aligned 3D Gaussian parameters. Extensive experi-

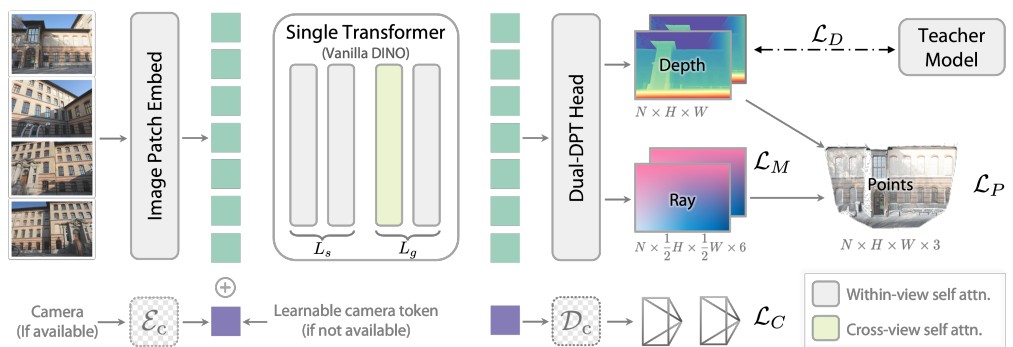

Figure 2: **Pipeline of Depth Anything 3.** Depth Anything 3 employs a single transformer (vanilla DINOv2 model) without any architectural modifications. To enable cross-view reasoning, an input-adaptive cross-view self-attention mechanism is introduced. A dual-DPT head is used to predict depth and ray maps from visual tokens. Camera parameters, if available, are encoded as camera tokens and concatenated with patch tokens, participating in all attention operations.

ments yield two key findings: 1) fine-tuning a geometry foundation model for NVS substantially outperforms highly specialized task-specific models (Xu et al., 2025c); 2) enhanced geometric reconstruction capability directly correlates with improved FF-NVS performance, establishing Depth Anything 3 as the optimal backbone for this task.

## 2 DEPTH ANYTHING 3

We tackle the recovery of consistent 3D geometry from diverse visual inputs—single image, multi-view collections, or videos—and optionally incorporate known camera poses when available.

### 2.1 FORMULATION

We denote the input as $\mathcal{I} = \{\mathbf{I}_i\}_{i=1}^{N_v}$ with each $\mathbf{I}_i \in \mathbb{R}^{H \times W \times 3}$. For $N_v = 1$ this is a monocular image, and for $N_v > 1$ it represents a video or multi-view set. Each image has depth $\mathbf{D}_i \in \mathbb{R}^{H \times W}$, camera pose $[\mathbf{R}_i \mid \mathbf{t}_i]$, and intrinsics $\mathbf{K}_i$. The camera can also be represented as $\mathbf{v}_i \in \mathbb{R}^9$ with translation $\mathbf{t}_i \in \mathbb{R}^3$, rotation quaternion $\mathbf{q}_i \in \mathbb{R}^4$, and FOV parameters $\mathbf{f}_i \in \mathbb{R}^2$. A pixel $\mathbf{p} = (u, v, 1)^\top$ projects to a 3D point $\mathbf{P} = (X, Y, Z, 1)^\top$ by

$$\mathbf{P} = \mathbf{R}_i\big(\mathbf{D}_i(u, v)\,\mathbf{K}_i^{-1}\mathbf{p}\big) + \mathbf{t}_i,$$

through which the underlying 3D visual space can be faithfully recovered.

**Depth-ray representation.** Predicting a valid rotation matrix $\mathbf{R}_i$ is challenging due to the orthogonality constraint. To avoid this, we represent camera pose implicitly with a per-pixel ray map, aligned with the input image and depth map. For each pixel $\mathbf{p}$, the camera ray $\mathbf{r} \in \mathbb{R}^6$ is defined by its origin $\mathbf{t} \in \mathbb{R}^3$ and direction $\mathbf{d} \in \mathbb{R}^3$: $\mathbf{r} = (\mathbf{t}, \mathbf{d})$. The direction is obtained by backprojecting $\mathbf{p}$ into the camera frame and rotating it to the world frame: $\mathbf{d} = \mathbf{R}\mathbf{K}^{-1}\mathbf{p}$. The dense ray map $\mathbf{M} \in \mathbb{R}^{H \times W \times 6}$ stores these parameters for all pixels. We do not normalize $\mathbf{d}$, so its magnitude preserves the projection scale. Thus, a 3D point in world coordinates is simply $\mathbf{P} = \mathbf{t} + \mathbf{D}(u, v) \cdot \mathbf{d}$. This formulation enables consistent point cloud generation by combining predicted depth and ray maps through element-wise operations.

**Minimal prediction targets.** Recent works aim to build unified models for diverse 3D tasks, often using multitask learning with different targets—for example, point maps alone (Wang et al., 2024b), or combinations of pose, local/global point maps, and depth (Wang et al., 2025a;b; Yang et al., 2025a). However, point maps inherently entangle depth and camera information into a single representation, which makes them less effective than disentangled depth predictions for geometry estimation (Wang et al., 2025a; Yang et al., 2025a). Consequently, prior works introduce additional depth heads alongside point maps, creating redundancy in the prediction targets. In contrast, our experiments (Tab. 5) show that a depth-ray representation forms a minimal yet sufficient disentangled target set for capturing both scene structure and camera motion, outperforming point map-based

alternatives. However, recovering camera pose from the ray map at inference is computationally costly. We address this by adding a camera head, $\mathcal{D}_C$, which has minimal computational overhead. This transformer operates on camera tokens to predict the field of view ($\mathbf{f} \in \mathbb{R}^2$), rotation as a quaternion ($\mathbf{q} \in \mathbb{R}^4$), and translation ($\mathbf{t} \in \mathbb{R}^3$). Since it processes only one token per view, the added computational cost is negligible.

## 2.2 ARCHITECTURE

We now detail the architecture of Depth Anything 3, which is illustrated in Fig. 2. The network is composed of three main components: a single transformer model as the backbone, an optional camera encoder for pose conditioning, and a Dual-DPT head for generating predictions.

**Single transformer backbone.** We use a Vision Transformer with $L$ blocks, pretrained on large-scale monocular image corpora (*e.g.*, DINOv2 Oquab et al. 2023). Cross-view reasoning is enabled without architectural changes via an input-adaptive self-attention, implemented by rearranging input tokens. We divide the transformer into two groups of sizes $L_s$ and $L_g$. The first $L_s$ layers apply self-attention within each image, while the subsequent $L_g$ layers alternate between cross-view and within-view attention, operating on all tokens jointly through tensor reordering. In practice, we set $L_s : L_g = 2 : 1$ with $L = L_s + L_g$. As shown in our ablation study in Tab. 6, this configuration provides the optimal trade-off between performance and efficiency compared to other arrangements. This design is input-adaptive: with a single image, the model naturally reduces to monocular depth estimation without extra cost.

**Camera condition injection.** To seamlessly handle both posed and unposed inputs, we prepend each view with a camera token $\mathbf{c}_i$. If camera parameters $(\mathbf{K}_i, \mathbf{R}_i, \mathbf{t}_i)$ are available, the token is obtained via a MLP $\mathcal{E}_c$: $\mathbf{c}_i = \mathcal{E}_c(\mathbf{f}_i, \mathbf{q}_i, \mathbf{t}_i)$. Otherwise, a shared learnable token $\mathbf{c}_l$ is used. Concatenated with patch tokens, these camera tokens participate in all attention operations, providing either explicit geometric context or a consistent learned placeholder.

**Dual-DPT head.** For the final prediction stage, we introduce a Dual-DPT head that jointly outputs dense depth and ray values, offering both strong and efficient (Tab. 5) results. Backbone features are first processed through shared reassembly modules, then split into two branches with distinct fusion layers for depth and rays, followed by separate output layers. Both branches thus operate on the same processed features, differing only in the fusion stage, which promotes interaction between tasks while avoiding redundant representations. Our model also outputs confidence map for depth following Wang et al. (2024b).

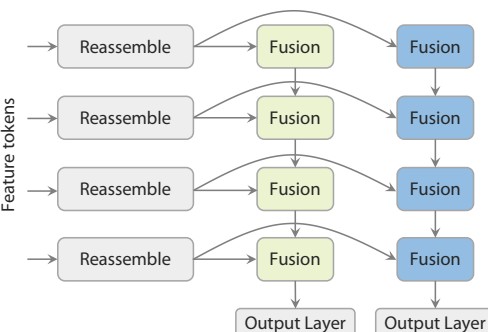

Figure 3: **Dual-DPT Head.** Two branchs share reassembly modules for better outputs alignment.

## 2.3 TRAINING

**Teacher-student learning paradigm.** Our training data comes from diverse sources, including real-world depth captures, 3D reconstructions, and synthetic datasets. Real-world depth is often noisy and incomplete (Fig. 7), limiting its supervisory value. To mitigate this, we train a monocular relative depth estimation "teacher" model solely on synthetic data to generate high-quality pseudo-labels. These pseudo-depth maps are aligned with the original sparse or noisy ground truth via RANSAC least squares, enhancing label detail and completeness while preserving geometric accuracy. We term this model Depth-Anything-3-Teacher, trained on a large synthetic corpus covering indoor, outdoor, object-centric, and diverse in-the-wild scenes to capture fine geometry. We detail our teacher design in the appendix.

**Training objectives.** Following the formulation in Sec. 2.1, our model $\mathcal{F}_\theta$ maps an input $\mathcal{I}$ to a set of outputs comprising a depth map $\hat{\mathbf{D}}$, a ray map $\hat{\mathbf{R}}$, and an optional camera pose $\hat{\mathbf{c}}$: $\mathcal{F}_\theta : \mathcal{I} \mapsto \{\hat{\mathbf{D}}, \hat{\mathbf{R}}, \hat{\mathbf{c}}\}$. The gray color indicates that $\hat{\mathbf{c}}$ is an optional output, included primarily for practical convenience. Prior to loss computation, all ground-truth signals are normalized by a common scale

factor. This scale is defined as the mean $\ell_2$ norm of the valid reprojected point maps $\mathbf{P}$, a step that ensures consistent magnitude across different modalities and stabilizes the training process. The overall training objective is defined as a weighted sum of several loss terms:

$$\mathcal{L} = \mathcal{L}_D(\hat{\mathbf{D}}, \mathbf{D}) + \mathcal{L}_M(\hat{\mathbf{R}}, \mathbf{M}) + \mathcal{L}_P(\hat{\mathbf{D}} \odot \mathbf{d} + \mathbf{t}, \mathbf{P}) + \beta\mathcal{L}_C(\hat{\mathbf{c}}, \mathbf{v}) + \alpha\mathcal{L}_{\text{grad}}(\hat{\mathbf{D}}, \mathbf{D}).$$

In practice, we set $\alpha = 1$ and $\beta = 1$. $\mathcal{L}_D$ is a confidence-aware loss following Wang et al. (2024b). $\mathcal{L}_{\text{grad}}$ is taken from Yang et al. (2024b) , penalizing the depth gradients. This loss preserves sharp edges while ensuring smoothness in planar regions. We detail the loss function in the appendix.

## 2.4 FINETUNING FOR FEED-FORWARD NOVEL VIEW SYNTHESIS

Inspired by human spatial intelligence, we believe that consistent depth estimation can greatly enhance downstream 3D vision tasks. We choose feed-forward novel view synthesis (FF-NVS) as the demonstration task, given its growing attention driven by advances in neural 3D representations and its relevance to numerous applications. Adhere to the minimal modeling strategy, we perform FF-NVS by fine-tuning with an added DPT head (GS-DPT) to infer pixel-aligned 3D Gaussians.

**GS-DPT Head.** Given visual tokens for each view extracted via our single transformer backbone (Sec. 2.2), GS-DPT predicts the camera-space 3D Gaussian parameters $\{\sigma_i, \mathbf{q}_i, \mathbf{s}_i, \mathbf{c}_i\}_{i=1}^{H \times W}$, where $\sigma_i, \mathbf{q}_i \in \mathbb{H}, \mathbf{s}_i \in \mathbb{R}^3, \mathbf{c}_i \in \mathbb{R}^3$ denote the opacity, rotation quaternion, scale, and RGB color of the $i$-th 3D Gaussian, respectively. Among them, $\sigma_i$ is predicted by the confidence head, while others are predicted by the main GS-DPT head. The estimated depth is unprojected to world coordinates to obtain the global positions $\mathbf{P}_i \in \mathbb{R}^3$ of the 3D Gaussians. These primitives are then rasterized to synthesize novel views from given camera poses. We detail our loss functions in the appendix.

## 3 VISUAL GEOMETRY BENCHMARK

We further introduce a visual geometry benchmark to assess geometry prediction models. It directly evaluates pose accuracy, depth via reconstruction accuracy and visual rendering quality.

**Pose accuracy.** Our benchmark covers 5 datasets: HiRoom (an internal high-fidelity room dataset), ETH3D (Schops et al., 2017), DTU (Aanæs et al., 2016), 7Scenes (Shotton et al., 2013), and ScanNet++ (Yeshwanth et al., 2023), containing 29, 11, 22, 7, and 20 scenes, respectively. These span object-centric to indoor and outdoor. **HiRoom and the benchmark will be released publicly.** ScanNet++ is not a zero-shot dataset, as it has been widely used for training since DUSt3R. Although comparisons are biased, we retain it for completeness since subsequent methods also adopt it. We report **Auc3** and **Auc30**, which measure relative rotation and translation score (higher is better).

**Geometry accuracy.** Using the same datasets, we assess depth accuracy via reconstruction. Unlike Wang et al. (2025a), which aligns predicted depths to ground truth with scale and shift and then reconstructs the scene with ground-truth poses, we reconstruct using both predicted poses and depths. The resulting point cloud is aligned to ground truth by applying evo (Umeyama, 2002) to match predicted poses with ground-truth poses. We report F-Score for all datasets except Chamfer Distance for DTU, following a prior work (Yu et al., 2022).

**Visual rendering quality.** We evaluate visual rendering quality on diverse large-scale scenes. We introduce a new NVS benchmark built from three datasets, including DL3DV (Ling et al., 2024) with 140 scenes, Tanks and Temples (Knapitsch et al., 2017a) with 6, and MegaDepth (Li and Snavely, 2018) with 19, each spanning around 300 sampled frames. Ground truth camera poses, estimated with COLMAP, are used directly to ensure accurate and fair comparison across diverse models. We report PSNR, SSIM, and LPIPS metrics on rendered novel views using given camera poses.

## 4 EXPERIMENTS

Training datasets, baselines, implementation details and more ablations are provided in the appendix.

Table 1: **Comparisons with SOTA methods on pose accuracy.** We report both Auc3 ↑ and Auc30 ↑ metrics. The top-3 results are highlighted as first , second , and third .

| Methods | HiRoom | | ETH3D | | DTU | | 7Scenes | | ScanNet++ | |
|---|---|---|---|---|---|---|---|---|---|---|
| | Auc3 | Auc30 | Auc3 | Auc30 | Auc3 | Auc30 | Auc3 | Auc30 | Auc3 | Auc30 |
| Colmap | 13.0 | 19.0 | 4.75 | 41.3 | 87.2 | 87.4 | 22.3 | 62.0 | 13.3 | 19.9 |
| Glomap | 31.9 | 42.6 | 8.37 | 21.7 | 96.8 | 96.9 | 24.1 | 85.0 | 20.8 | 42.9 |
| DUSt3R | 17.6 | 54.3 | 4.30 | 27.3 | 4.00 | 74.3 | 6.90 | 61.6 | 8.10 | 33.9 |
| Fast3R | 25.9 | 77.0 | 8.10 | 44.4 | 9.50 | 79.1 | 19.0 | 78.6 | 17.9 | 72.5 |
| MapAnything | 17.9 | 82.8 | 19.2 | 77.4 | 6.50 | 72.7 | 12.6 | 79.7 | 20.2 | 84.1 |
| Pi3 | 67.0 | 94.8 | 35.2 | 87.3 | 62.5 | 94.9 | 25.5 | 86.3 | 50.7 | 92.1 |
| VGGT | 49.1 | 88.0 | 26.3 | 80.8 | 79.2 | 99.8 | 23.9 | 85.0 | 62.6 | 95.1 |
| DA3-Giant | 81.7 | 96.4 | 39.3 | 90.6 | 85.6 | 94.9 | 29.2 | 86.8 | 83.2 | 98.0 |
| DA3-Large | 37.9 | 84.5 | 19.0 | 81.7 | 58.4 | 95.3 | 25.1 | 85.4 | 46.9 | 92.1 |
| DA3-Base | 12.8 | 79.8 | 13.6 | 74.0 | 31.4 | 90.8 | 17.2 | 81.1 | 16.2 | 77.5 |
| DA3-Small | 3.40 | 64.6 | 4.89 | 51.9 | 9.46 | 82.2 | 6.19 | 71.8 | 2.86 | 51.8 |

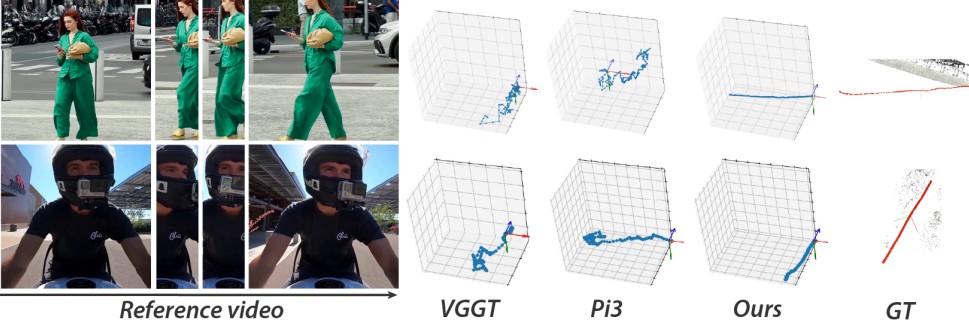

Figure 4: **Comparisons of pose estimation quality.** Camera trajectories for two videos are shown. Ground-truth trajectories are derived using COLMAP on images with dynamic objects masked.

## 4.1 VISUAL GEOMETRY ESTIMATION

**Pose estimation.** As shown in Tab. 1 and Fig. 4, we compare our method against both classical SfM pipelines (COLMAP Schönberger and Frahm (2016), GLOMAP Pan et al. (2024b)) and feed-forward methods (Wang et al., 2024b; 2025a; Yang et al., 2025a; Wang et al., 2025d; Keetha et al., 2025) using Auc3 and Auc30 metrics across five datasets.

Classical methods excel on dense, well-textured scenes like DTU, where GLOMAP achieves top performance (Auc3: 96.8). However, they struggle significantly on challenging scenarios with sparse views, textureless regions, or dynamic content. For instance, on HiRoom, COLMAP achieves only 13.0 Auc3 compared to 81.7 from our method. Similarly, on ScanNet++, COLMAP obtains 13.3 Auc3 versus 83.2 from ours.

Our DA3-Giant model establishes new SOTA results across nearly all metrics, with particularly strong performance on challenging datasets. On Auc3, our model delivers at least an **8%** relative improvement over all feed-forward competitors, and on ScanNet++ it achieves a **33%** relative gain over the second-best feed-forward model. To qualitatively assess pose quality, we visualize predicted camera trajectories on two in-the-wild dynamic videos in Fig. 4. Our trajectories are smooth and closely align with the ground truth, whereas VGGT and Pi3 exhibit substantially noisier paths.

**Geometry estimation.** As shown in Tab. 2, we compare our method against both classical structure-from-motion pipelines (COLMAP Schönberger and Frahm (2016), GLOMAP Pan et al. (2024b)) and state-of-the-art feed-forward reconstruction methods under two distinct conditions: a pose-free setting where camera parameters are unavailable, and a pose-conditioned setting where they are known.

Classical methods perform competitively on extremely dense datasets like DTU, where GLOMAP+PatchMatchStereo achieves 1.62 mm chamfer distance. However, their performance degrades significantly on datasets that are even slightly sparser. For example, on ETH3D, COLMAP+PM achieves an F-score of only 20.7 (ours: 74.4), and on HiRoom, it drops to 16.8

Table 2: **Comparisons with SOTA methods on reconstruction accuracy.** For all datasets except DTU, we report the F-Score (**F1** ↑). For DTU, we report the chamfer distance (**CD** ↓, unit: mm). w/o p. and w/ p. denote without pose and with pose, indicating whether ground-truth camera poses are provided for reconstruction. The top-3 results are highlighted as first , second , and third .

| Methods | HiRoom | | ETH3D | | DTU | | 7Scenes | | ScanNet++ | |
|---|---|---|---|---|---|---|---|---|---|---|
| | w/o p. | w/ p. | w/o p. | w/ p. | w/o p. | w/ p. | w/o p. | w/ p. | w/o p. | w/ p. |
| Colmap+PM | 16.8 | 21.9 | 20.7 | 25.7 | 1.67 | 1.64 | 40.0 | 37.3 | 15.7 | 19.6 |
| Glomap+PM | 30.7 | 41.1 | 24.1 | 36.3 | 1.62 | 1.59 | 43.9 | 47.6 | 16.5 | 22.4 |
| DUSt3R | 30.1 | 39.5 | 19.7 | 18.8 | 7.60 | 7.97 | 26.6 | 39.8 | 18.9 | 27.3 |
| Fast3R | 40.7 | 48.2 | 38.5 | 50.3 | 6.88 | 8.20 | 41.0 | 49.8 | 37.1 | 53.7 |
| MapAnything | 32.4 | 69.2 | 54.8 | 71.9 | 7.91 | 3.97 | 44.8 | 55.2 | 39.4 | 71.3 |
| Pi3 | 75.8 | 85.0 | 72.7 | 80.6 | 3.28 | 1.72 | 44.2 | 57.5 | 63.1 | 73.3 |
| VGGT | 56.7 | 70.2 | 57.2 | 66.7 | 2.05 | 1.44 | 47.9 | 51.4 | 66.4 | 70.7 |
| DA3-Giant | 89.3 | 95.2 | 74.4 | 85.8 | 1.92 | 0.91 | 52.0 | 52.3 | 76.4 | 79.2 |
| DA3-Large | 48.2 | 85.7 | 57.3 | 79.1 | 3.45 | 2.48 | 48.7 | 48.7 | 58.9 | 72.9 |
| DA3-Base | 18.6 | 71.7 | 52.8 | 66.6 | 5.14 | 1.99 | 37.8 | 47.2 | 39.7 | 66.3 |
| DA3-Small | 12.9 | 43.1 | 39.4 | 58.2 | 5.12 | 4.05 | 30.8 | 39.5 | 24.2 | 45.7 |

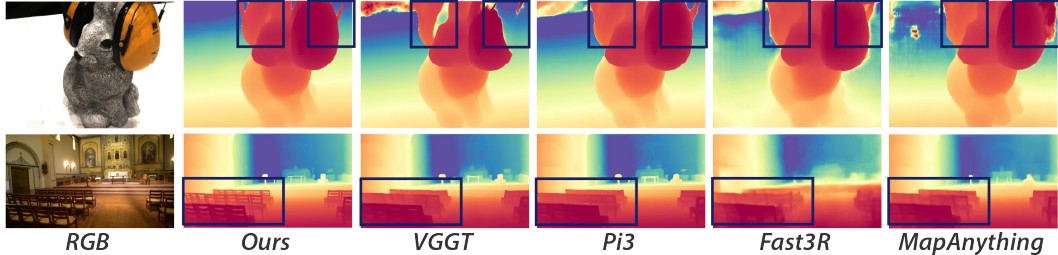

| RGB | Ours | VGGT | Pi3 | Fast3R | MapAnything |

Figure 5: **Comparisons of depth quality.** Compared with other methods, our depth maps exhibit finer structural detail and higher semantic correctness across diverse scenes.

(ours: 89.3). This highlights the limitations of classical methods in handling sparse-view scenarios, where correspondence matching becomes unreliable.

In contrast, our DA3-Gaint establishes a new SOTA across nearly all scenarios, outperforming all feed-forward competitors in all five pose-free settings. On average, DA3-Gaint achieves a relative improvement of 23.6% over VGGT and 16.7% over Pi3. Fig. 5 and Fig. 6 visualize our predicted depth and recovered point clouds. The results are not only clean, accurate, and complete, but also preserve fine-grained geometric details, clearly demonstrating superiority over other methods.

Even more notably, our much smaller DA3-Large (0.30B parameters) demonstrates remarkable efficiency. Despite being 3× smaller than VGGT (0.90B parameters), it surpasses VGGT in five out of the ten settings, with particularly strong performance on ETH3D.

When camera poses are available, both our method and MapAnything can exploit them for improved results, and other methods also benefit from ground-truth pose fusion. Our model shows clear gains on most datasets except 7Scenes, where the limited video setting already saturates performance and reduces the benefit of pose conditioning. Notably, with pose conditioning, performance gains from scaling model size are smaller than in pose-free models, **indicating that pose estimation scales more strongly than depth estimation and requires larger models to fully realize improvements.**

Monocular depth accuracy also reflects geometry quality. As shown in Tab. 3, on the standard monocular depth benchmarks reported in Yang et al. (2024b), our model outperforms VGGT and Depth Anything 2. For reference, we also include the results of our teacher model.

Table 3: **Monocular depth comparisons.** $\delta_1$ ↑

| Method | KITTI | NYU | SINTEL | ETH3D | DIODE | Rank |
|---|---|---|---|---|---|---|
| DA2 | 94.6 | 97.9 | 77.2 | 86.5 | 95.2 | 2.60 |
| VGGT | 91.7 | 97.9 | 67.9 | 97.5 | 95.3 | 3.75 |
| DA3 | 95.3 | 97.4 | 75.5 | 98.6 | 95.4 | 2.20 |
| Teacher | 97.2 | 97.9 | 81.4 | 99.8 | 96.6 | 1.00 |

**Visual rendering.** To fairly evaluate feed-forward novel view synthesis (FF-NVS), we compare against three recent 3DGS models—pixelSplat (Charatan et al., 2024), MVSplat (Chen et al., 2024), and DepthSplat (Xu

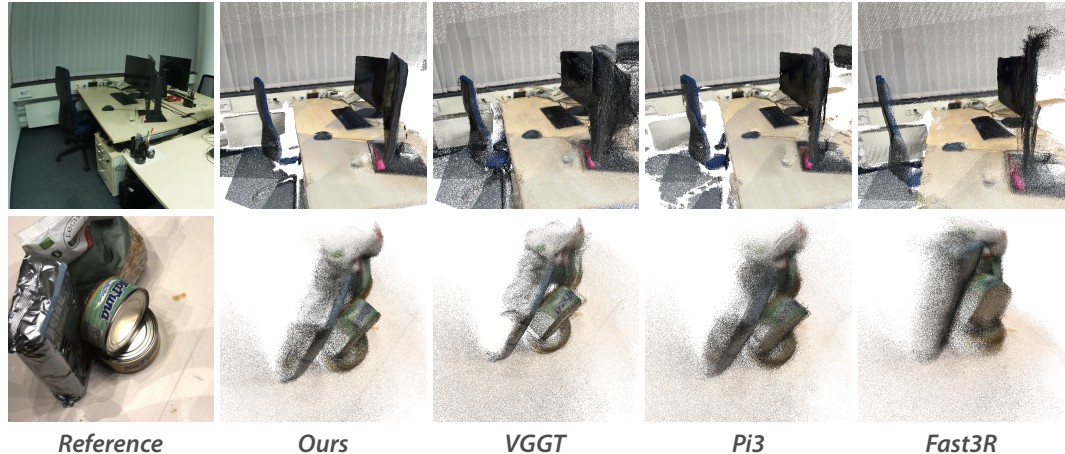

Reference   Ours   VGGT   Pi3   Fast3R

Figure 6: **Comparisons of point cloud quality.** Our model produces point clouds that are more geometrically regular and substantially less noisy than those generated by other methods.

Table 5: **Ablations of prediction-target combinations.** Note that all experiments in this table do not have camera condition token. The **best** and second best are highlighted.

| Methods | Avg | | HiRoom | | ETH3D | | DTU | | 7Scenes | | ScanNet++ | |
|---|---|---|---|---|---|---|---|---|---|---|---|---|
| | Auc3 | F1 | Auc3 | F1 | Auc3 | F1 | Auc3 | CD | Auc3 | F1 | Auc3 | F1 |
| pcd | 31.6 | 51.5 | 35.9 | 42.7 | 20.3 | 64.6 | 49.1 | 4.219 | 21.7 | 45.5 | 30.8 | 53.3 |
| depth + cam | 14.1 | 40.2 | 10.8 | 16.5 | 9.9 | 48.0 | 23.3 | 5.316 | 13.0 | 38.5 | 13.3 | 41.0 |
| depth + cam + pcd | 22.6 | 42.9 | 9.1 | 12.8 | 19.0 | 60.4 | 42.3 | 4.918 | 20.8 | 43.4 | 22.0 | 43.0 |
| depth + ray | 36.0 | 56.4 | 48.7 | 60.3 | **25.5** | **65.4** | 46.5 | 3.919 | 24.0 | 46.5 | **35.5** | 53.4 |
| depth + ray + pcd | **36.4** | **56.5** | **52.3** | **61.4** | 22.6 | 64.2 | 43.0 | 4.158 | **27.3** | **48.5** | 36.7 | 51.8 |
| depth + ray + cam | 35.1 | 51.7 | 37.2 | 45.4 | 22.3 | 59.4 | **56.3** | **3.066** | 25.7 | 45.6 | 34.1 | **56.5** |

et al., 2025c)—and further test alternative frameworks by replacing our geometry backbone with Fast3R (Yang et al., 2025b), MV-DUSt3R (Tang et al., 2025), and VGGT (Wang et al., 2025a). All models are trained on DL3DV-10K training set under a unified protocol and evaluated on our benchmark (Sec. 3).

As shown in Tab. 4, all models perform substantially better on DL3DV than on the other datasets, suggesting that 3DGS-based NVS is sensitive to trajectory and pose distributions standardized by DL3DV, rather than scene content. Comparing the two groups, geometry-model-based frameworks consistently outperform specialized feed-forward models, demonstrating that a simple backbone plus DPT head can surpass complex task-specific designs. The advantage stems from large-scale pretraining, which enables better generalization and scalability than approaches relying on epipolar transformers, cost volumes, or cascaded modules. Within this group, NVS performance correlates with geometry estimation capability, making DA3 the strongest backbone. Looking forward, we expect FF-NVS can be effectively addressed with simple architectures leveraging pretrained geometry backbones, and that the strong spatial understanding of DA3 will benefit other 3D vision tasks.

Table 4: **Comparisons with SOTA methods on NVS task.** We report NVS comparsions with exisiting feed-forward 3DGS models and counterparts using other backbones.

| Methods | In-domain Dataset | | Out-of-domain Datasets | | | |
|---|---|---|---|---|---|---|
| | DL3DV | | Tanks&Temples | | MegaDepth | |
| | PSNR↑ | LPIPS↓ | PSNR↑ | LPIPS↓ | PSNR↑ | LPIPS↓ |
| pixelSplat | 16.55 | 0.480 | 13.81 | 0.558 | 13.87 | 0.561 |
| MVSplat | 18.13 | 0.393 | 14.81 | 0.508 | 14.67 | 0.533 |
| DepthSplat | 19.24 | 0.322 | 15.80 | 0.418 | 15.90 | 0.450 |
| Fast3R | 19.30 | 0.320 | 16.24 | 0.409 | 16.43 | 0.421 |
| MV-DUSt3R | 20.01 | 0.294 | 17.04 | 0.370 | 16.20 | 0.437 |
| VGGT | 20.96 | 0.253 | 17.18 | 0.347 | 16.45 | 0.417 |
| DA3 | **21.33** | **0.241** | **18.10** | **0.311** | **17.89** | **0.351** |

Table 6: **Ablations for single transformer.** We evaluate three architectural designs with comparable model sizes. The **best** and second best are highlighted.

| Methods | HiRoom | | ETH3D | | DTU | | 7Scenes | | ScanNet++ | |
|---|---|---|---|---|---|---|---|---|---|---|
| | Auc3↑ | F1↑ | Auc3↑ | F1↑ | Auc3↑ | CD↓ | Auc3↑ | F1↑ | Auc3↑ | F1↑ |
| a. Proposed Arch. | **39.2** | **47.0** | **21.0** | **55.4** | **45.8** | **3.82** | **26.2** | 47.6 | **30.3** | **51.1** |
| b. VGGT Style | 3.72 | 14.5 | 2.31 | 27.4 | 1.38 | 6.93 | 0.97 | 21.4 | 2.03 | 12.2 |
| c. Full Alt. | 24.7 | 29.3 | 13.1 | 51.9 | 44.6 | 4.23 | 21.1 | **48.6** | 27.7 | 47.5 |

## 4.2 SUFFICIENCY OF THE DEPTH-RAY REPRESENTATION

To validate our depth-ray representation, we compare different prediction combinations in Tab. 5. All models use a ViT-L backbone with identical training settings (view size: 10, batch size: 128, steps: 120k). We evaluate four prediction heads: 1) **depth** for dense depth maps; 2) **pcd** for direct 3D point clouds; 3) **cam** for 9-DoF camera pose $\mathbf{c} = (\mathbf{t}, \mathbf{q}, \mathbf{f})$; and 4) our proposed **ray**, predicting per-pixel ray maps (Sec. 2.1). The **ray** head uses a Dual-DPT architecture, while **pcd** uses a separate DPT head. For models without **pcd**, point clouds are obtained by combining **depth** with camera parameters from **ray** or **cam**.

**Point cloud prediction is insufficient.** Directly predicting point clouds (**pcd**) performs poorly (Avg: 31.6 Auc3, 51.5 F1), as the point cloud representation inherently lacks the geometric structure needed for accurate camera pose estimation. Adding point cloud supervision to the depth-ray model (**depth + ray + pcd**) yields marginal improvements on some datasets but actually degrades performance on others, achieving only 36.4 Auc3 and 56.5 F1 on average.

**Depth-ray representation excels.** Our minimal **depth + ray** configuration achieves strong and balanced performance (36.0 Auc3, 56.4 F1), significantly outperforming **depth + cam** (14.1 Auc3, 40.2 F1) by over **155%** relative gain in Auc3. Notably, **depth + ray** also surpasses **depth + ray + pcd** despite being simpler, demonstrating that the point cloud head introduces unnecessary complexity without consistent benefits. The depth-ray formulation effectively captures both metric depth and camera geometry in a unified framework.

We adopt **depth + ray + cam** as our final configuration. Since the auxiliary **cam** head incurs negligible computational overhead (∼0.1% of backbone cost) and is significantly faster than extracting camera parameters from ray maps through optimization (0.46ms v.s 8.60ms on an A100 GPU), we include it at no practical cost.

## 4.3 SUFFICIENCY OF A SINGLE PLAIN TRANSFORMER

We compare a standard ViT-L backbone with a VGGT-style architecture that stacks two distinct transformers, tripling the block count. For fair capacity comparison, the VGGT-style model uses smaller ViT-B backbones, yielding a similar parameter size to our ViT-L. Our backbone supports two attention strategies: **Full Alt.**, which alternates cross-view/within-view attention in all layers ($L = L_g$), and our default partial alternation. As shown in Table 6, the VGGT-style model drops to 79.8% of our baseline performance, confirming the superiority of a single-transformer design at similar scale. We attribute this gap to full pretraining of our backbone versus two-thirds untrained blocks in VGGT. Moreover, the **Full Alt.** variant degrades across nearly all metrics—except F1 on 7Scenes—indicating that partial alternation is the more effective and robust strategy.

## 5 RELATED WORK

**Multi-view visual geometry estimation.** Traditional systems (Schönberger and Frahm, 2016; Schönberger et al., 2016) decompose reconstruction into feature detection and matching, robust relative pose estimation, incremental or global SfM with bundle adjustment, and dense multi-view stereo for per-view depth and fused point clouds. These methods remain strong on well-textured scenes, but their modularity and brittle correspondences complicate robustness under low texture, specularities, or large viewpoint changes. Early learning methods injected robustness at the component level: learned detectors (DeTone et al., 2018), descriptors for matching (Dusmanu et al., 2019), and differentiable optimization layers that expose pose/depth updates to gradient flow (He

et al., 2024; Guo et al., 2025; Pan et al., 2024a). On the dense side, cost-volume networks (Yao et al., 2018; Xu et al., 2023; 2025b) for MVS replaced hand-crafted regularization with 3D CNNs, improving depth accuracy especially at large baselines and thin structures compared with classical PatchMatch. Early end-to-end approaches (Teed and Deng, 2018; Wang et al., 2024a) moved beyond modular SfM/MVS pipelines by directly regressing camera poses and per-image depths from pairs of images. These approaches reduced engineering complexity and demonstrated the feasibility of learned joint depth pose estimation, but they often struggled with scalability, generalization, and handling arbitrary input cardinalities.

A turning point came with DUSt3R (Wang et al., 2024b), which leveraged transformers to directly predict point map between two views and compute both depth and relative pose in a purely feed-forward manner. This work laid the foundation for subsequent transformer-based methods aiming to unify multi-view geometry estimation at scale. Follow-up models extended this paradigm with multi-view inputs (Yang et al., 2025a; Wang et al., 2025b; Tang et al., 2025), video input (Zhang et al., 2025a; Wang et al., 2025b; Murai et al., 2025), robust correspondence modeling (Leroy et al., 2024), camera parameter injection (Jang et al., 2025; Keetha et al., 2025), view synthesis (Zhang et al., 2025b), and multi-modal prior prompting (Liu et al., 2025). Among these, Wang et al. (2025a) push accuracy to a new level through large-scale training, a multi-stage architecture, and redundancy in design. Parallel to this transformer-centric evolution, a complementary line of works (Zhang et al., 2024; Lu et al., 2025) explores diffusion models for ray-based camera estimation. These methods perform inference via iterative sampling, incurring higher computational cost than single-pass predictors. In contrast, we focus on a minimal modeling strategy built around a single, simple transformer.

**Monocular depth estimation.** Early monocular depth estimation methods relied on fully supervised learning on single-domain datasets, which often produced models specialized to either indoor rooms (Silberman et al., 2012) or outdoor driving scenes (Geiger et al., 2013). These early deep models achieved good accuracy within their training domain but struggled to generalize to novel environments, highlighting the challenge of cross-domain depth prediction. Modern generalist approaches (Yang et al., 2024a;b; Wang et al., 2025c; Bochkovskii et al., 2024; Yin et al., 2023; Ke et al., 2024) exemplify this trend by leveraging massive multi-dataset training and advanced architectures like vision transformers (Ranftl et al., 2021) or DiT (Peebles and Xie, 2023). Trained on millions of images, they learn broad visual cues and incorporate techniques such as affine-invariant depth normalization. Recent efforts further push toward pixel-perfect and arbitrary-resolution depth estimation (Xu et al., 2025a; 2026; Yu et al., 2026), producing fine-grained, flying-pixel-free predictions through pixel-space diffusion or neural implicit representations. In contrast, our method is primarily designed for a unified visual geometry estimation task, yet it still demonstrates competitive monocular depth performance.

## 6  CONCLUSION AND DISCUSSION

Depth Anything 3 shows that a plain transformer, trained on depth-and-ray targets with teacher–student supervision, can unify any-view geometry without ornate architectures. Scale-aware depth, per-pixel rays, and adaptive cross-view attention let the model inherit strong pretrained features while remaining lightweight and easy to extend. On the proposed visual geometry benchmark the approach sets new pose and reconstruction records, with both giant and compact variants surpassing prior models, while the same backbone powers efficient feed-forward novel view synthesis model.

We view Depth Anything 3 as a step toward versatile 3D foundation models. Future work can extend its reasoning to dynamic scenes, integrate language and interaction cues, and explore larger-scale pretraining to close the loop between geometry understanding and actionable world models. We hope the model and dataset releases, benchmark, and simple modeling principles offered here catalyze broader research on general-purpose 3D perception.

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

# A  METHOD DETAILS

## A.1  DERIVING CAMERA PARAMETERS FROM THE RAY MAP

Given an input image $I \in \mathbb{R}^{H \times W \times 3}$, the corresponding ray map is denoted by $\mathbf{M} \in \mathbb{R}^{H \times W \times 6}$. This map comprises per-pixel ray origins, stored in the first three channels ($\mathbf{M}(:,:,:3)$), and ray directions, stored in the last three ($\mathbf{M}(:,:,3:)$).

First, the camera center $\mathbf{t}_c$ is estimated by averaging the per-pixel ray origin vectors:

$$\mathbf{t}_c = \frac{1}{H \times W} \sum_{h=1}^{H} \sum_{w=1}^{W} \mathbf{M}(h, w, :3). \tag{1}$$

To estimate the rotation $\mathbf{R}$ and intrinsics $\mathbf{K}$, we formulate the problem as finding a homography $\mathbf{H}$. We begin by defining an "identity" camera with an identity intrinsics matrix, $\mathbf{K}_I = \mathbf{I}$. For a given pixel $\mathbf{p}$, its corresponding ray direction in this canonical camera's coordinate system is simply $\mathbf{d}_I = \mathbf{K}_I^{-1} \mathbf{p} = \mathbf{p}$. The transformation from this canonical ray $\mathbf{d}_I$ to the ray direction $\mathbf{d}_{\text{cam}}$ in the target camera's coordinate system is given by $\mathbf{d}_{\text{cam}} = \mathbf{K}\mathbf{R}\mathbf{d}_I$. This establishes a direct homography relationship, $\mathbf{H} = \mathbf{K}\mathbf{R}$, between the two sets of rays. We can then solve for this homography by minimizing the geometric error between the transformed canonical rays and a set of pre-computed target rays, $\mathbf{M}(h, w, 3:)$. This leads to the following optimization problem:

$$\mathbf{H}^* = \arg \min_{||\mathbf{H}||=1} \sum_{h=1}^{H} \sum_{w=1}^{W} ||\mathbf{H}\mathbf{p}_{h,w} \times \mathbf{M}(h, w, 3:)||. \tag{2}$$

This is a standard least-squares problem that can be efficiently solved using the Direct Linear Transform (DLT) algorithm (Abdel-Aziz et al., 2015). Once the optimal homography $\mathbf{H}^*$ is found, we recover the camera parameters. Since the intrinsics matrix $\mathbf{K}$ is upper-triangular and the rotation matrix $\mathbf{R}$ is orthonormal, we can uniquely decompose $\mathbf{H}^*$ using RQ decomposition to obtain $\mathbf{K}, \mathbf{R}$.

## A.2  CAMERA HEAD VS. RAY-BASED POSE ESTIMATION

Our model includes both a ray prediction head and an optional camera head for pose estimation. The ray prediction is essential for achieving high pose accuracy during training, as it provides dense per-pixel geometric supervision. The camera head, while optional, offers a significant advantage for inference speed.

**Computational efficiency.** Directly predicting camera parameters from the camera head is substantially faster than solving for pose from the ray map. Specifically, direct camera head prediction takes only 0.46 ms, whereas solving pose from the ray map using the DLT-based optimization described in Sec. A.1 requires approximately 8.60 ms (measured on an A100 GPU, averaged over a 49-image scene; for reference, the feature forward pass with image token attention takes 43 ms). This represents an 18.7× speedup.

**Accuracy trade-off.** While the camera head provides faster inference, ray-based pose estimation achieves slightly higher accuracy. As shown in Tab. 7, extracting pose from ray maps yields an average Auc3 of 68.0 compared to 63.8 from the camera head. However, the camera head still delivers strong performance across all datasets, making it a practical choice for applications requiring real-time inference.

In our final model, we include both heads: the ray head provides dense geometric predictions essential for training and high-accuracy applications, while the camera head enables faster inference when speed is prioritized. Since the camera head processes only one token per view, its computational overhead is negligible (∼0.1% of backbone cost).

## A.3  PIXEL-WISE RAY ORIGIN PREDICTION

In our ray map representation, each pixel predicts both a ray origin and a ray direction. An alternative design would be to predict a single global ray origin (camera center) using an MLP. We experimentally observed that predicted per-pixel ray origins exhibit extremely small variance and

Table 7: **Comparison between camera head and ray-based pose estimation for DA3-Giant.**
Time is measured for pose extraction per view on an A100 GPU.

| Method | Time (ms) | Avg | HiRoom | ETH3D | DTU | 7Scenes | ScanNet++ |
|---|---|---|---|---|---|---|---|
| Camera head | 0.46 | 63.8 | 81.7 | 39.3 | 85.6 | 29.2 | 83.2 |
| Ray-based | 8.60 | 68.0 | 88.6 | 42.4 | 89.3 | 29.6 | 89.9 |

Table 8: **Ablation study on ray origin prediction.** We compare per-pixel ray origin prediction
(*depth + ray + cam*) with single MLP-based global prediction (*depth + ray-st + cam*).

| Method | Avg | HiRoom | ETH3D | DTU | 7Scenes | ScanNet++ |
|---|---|---|---|---|---|---|
| depth + ray + cam | 35.1 | 37.2 | 22.3 | 56.3 | 25.7 | 34.1 |
| depth + ray-st + cam | 32.2 | 28.7 | 22.7 | 48.8 | 27.0 | 33.9 |

are essentially constant across pixels. As shown in Tab. 8, replacing per-pixel prediction with a single MLP-based prediction (*depth + ray-st + cam*) leads to a performance drop (average Auc3: 35.1 $\rightarrow$ 32.2), suggesting that the dense formulation provides better performance.

## A.4 DETAILS OF TRAINING OBJECTIVE

We define the loss terms in Equation 2.3 as follows.

$$\mathcal{L}_D(\hat{\mathbf{D}}, \mathbf{D}; D_c) = \frac{1}{Z_\Omega} \sum_{p \in \Omega} m_p \left( D_{c,p} \left| \hat{\mathbf{D}}_p - \mathbf{D}_p \right| - \lambda_c \log D_{c,p} \right),$$

where $D_{c,p}$ denotes the confidence of depth $D_p$. All loss terms are based on the $\ell_1$ norm, with weights set to $\alpha = 1$ and $\beta = 1$. The gradient loss, $\mathcal{L}_{\text{grad}}$, penalizes the depth gradients:

$$\mathcal{L}_{\text{grad}}(\hat{\mathbf{D}}, \mathbf{D}) = ||\nabla_x \hat{\mathbf{D}} - \nabla_x \mathbf{D}||_1 + ||\nabla_y \hat{\mathbf{D}} - \nabla_y \mathbf{D}||_1, \tag{3}$$

where $\nabla_x$ and $\nabla_y$ are the horizontal and vertical finite difference operators. This loss preserves sharp edges while ensuring smoothness in planar regions.

## A.5 DEPTH ANYTHING 3 TEACHER MODEL

As shown in Fig. 7, the real-world datasets are of poor quality, thus we train the teacher model exclusively on synthetic data to provide supervision for real-world data.

**Data scaling.** Following DA2, we train the teacher model exclusively on synthetic data to achieve finer geometric detail. However, the synthetic datasets used in DA2 are relatively limited. In DA3, we substantially expand the training corpus to include: Hypersim (Roberts et al., 2021), TartanAir (Wang et al., 2020), IRS (Wang et al., 2019), vKITTI2 (Cabon et al., 2020), BlendedMVS (Yao et al., 2020),

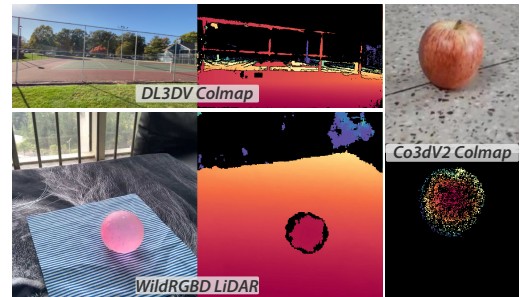

Figure 7: **Poor quality real-world datasets.** We show some examples of the poor quality real-world datasets.

SPRING (Mehl et al., 2023), MVSSynth (Huang et al., 2018), UnrealStereo4K (Zhang et al., 2018), GTA-SfM (Wang and Shen, 2020), TauAgent (Gil et al., 2021), KenBurns Niklaus et al. (2019), MatrixCity (Li et al., 2023), EDEN (Le et al., 2021), ReplicaGSO (Straub et al., 2019a), Urban-Syn (Gómez et al., 2025), PointOdyssey (Zheng et al., 2023), Structured3D (Zheng et al., 2020), Objaverse (Deitke et al., 2023), Trellis (Xiang et al., 2024), and OmniObject (Wu et al., 2023). This collection spans indoor, outdoor, object-centric, and diverse in-the-wild scenes, improving generalization of the teacher model.

**Depth representation.** Unlike DA2, which predicts scale–shift-invariant disparity, our teacher outputs scale–shift-invariant depth. Depth is preferable for downstream tasks, such as metric depth

estimation and multiview geometry, that directly operate in depth space rather than disparity. To address depth's reduced sensitivity for near-camera regions comparing to disparity, we predict exponential depth instead of linear depth, enhancing discrimination at small distances.

**Training objectives.** For geometric supervision, in addition to a standard depth-gradient loss, we adopt ROE alignment with the global–local loss introduced in Wang et al. (2025c). To further refine local geometry, we introduce a distance-weighted surface-normal loss. For each center pixel, we sample four neighboring points and compute unnormalized normals $n_i$. We then weight these normals by:

$$w_i = \sum_{j=0}^{4} \parallel n_j \parallel - \parallel n_i \parallel, \tag{4}$$

which downweights contributions from neighbors farther from the center, yielding a mean normal closer to the true local surface normal:

$$n_m = \sum_{i=0}^{4} w_i \frac{n_i}{\parallel n_i \parallel}, \tag{5}$$

The final normal loss is

$$\mathcal{L}_N = \mathcal{E}(\hat{n}_m, n_m) + \sum_{i=0}^{4} \mathcal{E}(\hat{n}_i, n_i) \tag{6}$$

where $\mathcal{E}$ denotes the angular error between normals. Ground truth is undefined in sky regions and in background areas of object-only datasets. To prevent these regions from degrading the depth prediction and to facilitate downstream use, we jointly predict a sky mask and an object mask aligned with the depth output, supervised with MSE loss. The overall training objective is

$$\mathcal{L}_T = \alpha \mathcal{L}_{\text{grad}} + \mathcal{L}_{\text{gl}} + \mathcal{L}_N + \mathcal{L}_{\text{sky}} + \mathcal{L}_{\text{obj}} \tag{7}$$

where $\alpha = 0.5$. Here, $\mathcal{L}_{\text{grad}}$, $\mathcal{L}_{\text{gl}}$, $\mathcal{L}_{\text{sky}}$, and $\mathcal{L}_{\text{obj}}$ denote the gradient loss, global–local loss, sky-mask loss, and object-mask loss, respectively.

### A.6 VISUAL RENDERING DETAILS.

The NVS model is fine-tuned with two training objectives, namely photometric loss (*i.e.*, $\mathcal{L}_{\text{MSE}}$ and $\mathcal{L}_{\text{LPIPS}}$) on rendered novel views and scale-shift-invariant depth loss $\mathcal{L}_D$ on the estimated depth of observed views, following the teacher–student learning paradigm (Sec. 2.3).

## B VISUAL GEOMETRY BENCHMARK

### B.1 MORE DETAILS ABOUT EVALUATION PIPELINE

**Pose estimation.** For each scene, we select all available images; if the total number exceeds the limit, we randomly sample 100 images using a fixed random seed. The selected images are then processed through a feed-forward model to generate consistent pose and depth estimations, after which the pose accuracy is computed.

**Geometry estimation.** For the same image set, we perform reconstruction using the predicted poses together with the predicted depths. To align the reconstructed point cloud with the ground-truth, we employ evo (Umeyama, 2002) to align the predicted poses to the ground-truth poses, obtaining a transformation that maps the reconstruction into the ground-truth coordinate system. To improve robustness, we adopt a RANSAC-based alignment procedure. Specifically, we repeatedly apply evo on randomly sampled pose subsets and evaluate each candidate transformation by counting the number of inlier poses, where inliers are defined as those with translation errors below the median of the overall pose deviations. The transformation with the largest inlier set is then chosen and applied to fuse the aligned predicted point cloud with the predicted depth maps by TSDF fusion. Finally, reconstruction quality is assessed by comparing the aligned reconstruction with the ground-truth point cloud using the metrics described in Sec. B.3.

**Visual rendering.** For each testing scene, the number of images typically ranges from 300 to 400 across all benchmark datasets. We sample one out of every 8 images as target novel views for evaluation. From the remaining viewpoints, we use COLMAP camera poses provided by each dataset and apply farthest point sampling, considering both camera translation and rotation distances, to select 12 images as input context views. For DL3DV, we use the official Benchmark set for testing. For Tanks and Temples, all Training Data scenes are included except `Courthouse`. For MegaDepth, we select scenes numbered from `5000` to `5018`, as these are most suitable for NVS.

## B.2 POSE METRICS

For assessing pose estimation, we follow the evaluation protocol introduced in Wang et al. (2025a; 2023) and report results using the AUC. This metric is derived from two components: Relative Rotation Accuracy (RRA) and Relative Translation Accuracy (RTA). RRA and RTA quantify the angular deviation in rotation and translation, respectively, between two images. Each error is compared against a set of thresholds to obtain accuracy values. AUC is then computed as the integral of the accuracy–threshold curve, where the curve is determined by the smaller of RRA and RTA at each threshold. To illustrate performance under different tolerance levels, we primarily report results at thresholds of 3 and 30.

## B.3 RECONSTRUTION METRICS

Let $\mathcal{G}$ denote the ground-truth point set and $\mathcal{R}$ the reconstructed point set under evaluation. We measure accuracy using $\text{dist}(\mathcal{R} \rightarrow \mathcal{G})$ and completeness using $\text{dist}(\mathcal{G} \rightarrow \mathcal{R})$ following Aanæs et al. (2016). The Chamfer Distance (CD) is then defined as the average of these two terms. Based on these distances, we define the precision and recall of the reconstruction $\mathcal{R}$ with respect to a distance threshold $d$. Precision is given by $\frac{1}{|\mathcal{R}|} \sum \left[ \text{dist}(\mathcal{R}_i \rightarrow \mathcal{G}) < d \right]$, and recall by $\frac{1}{|\mathcal{G}|} \sum \left[ \text{dist}(\mathcal{G}_i \rightarrow \mathcal{R}) < d \right]$, where $[\cdot]$ denotes the Iverson bracket Knapitsch et al. (2017b). To jointly capture both measures, we report the F1-score, computed as $\text{F1} = \frac{2 \times \text{precision} \times \text{recall}}{\text{precision} + \text{recall}}$.

## B.4 DATASETS

Our benchmark is built on five datasets: HiRoom, ETH3D (Schops et al., 2017), DTU (Aanæs et al., 2016), 7Scenes (Shotton et al., 2013), and ScanNet++ (Yeshwanth et al., 2023). Together, they cover diverse scenarios ranging from object-centric captures to complex indoor and outdoor environments, and are widely adopted in prior research. Below, we present more details about the dataset preparation process.

**HiRoom** is a Blender-rendered synthetic dataset comprising 30 indoor living scenes created by professional artists. We use a threshold $d$ of 0.05m for the F1 reconstruction metric calculation. For TSDF fusion, we set the parameters voxel size to 0.007m.

**ETH3D** provides high-resolution indoor and outdoor images with ground-truth depth from laser sensors. We aggregate the ground-truth depth maps with TSDF fusion for GT 3D shapes. We select 11 scenes: `courtyard`, `electro`, `kicker`, `pipes`, `relief`, `delivery area`, `facade`, `office`, `playground`, `relief 2`, `terrains`, for the benchmark. All frames are used in the evaluation. We use a threshold $d$ of 0.25 for the F1 reconstruction metric calculation. For TSDF fusion, we set the parameters voxel size to 0.039m.

**DTU** is an indoor dataset consisting of 124 different objects, each scene is recorded from 49 views. It provides ground-truth point clouds collected under well-controlled conditions. We evaluate models on the 22 evaluation scans of the DTU dataset following Yao et al. (2018). We adopt the RMBG 2.0 Zheng et al. (2024) to remove meaningless background pixels and use the default depth fusion strategy proposed in Zhang et al. (2023). All frames are used in the evaluation.

**7Scenes** is a challenging real-world dataset, consisting of low-resolution images with severe motion blurs for in-door scenes. We follow the implementation in Zhu et al. (2024) to fuse RGBD images with TSDF fusion and prepare ground-truth 3D shapes. We downsample the number of frames for each scene by 11 to faciliate evaluation. We use a threshold $d$ of 0.05m for the F1 reconstruction metric calculation. For TSDF fusion, we set the parameters voxel size to 0.007m.

**ScanNet++** is an extensive indoor dataset providing high-resolution images, depth maps from iPhone LiDAR, and high-resolution depth maps sampled from reconstructions of laser scans. We select 20 scenes for the benchmark. As depth maps from iPhone LiDAR lack of invalid ground-truth indicators, we use depth maps sampled from reconstructions of laser scans as ground-truth depth by default. We aggregate the ground-truth depth maps with TSDF fusion for GT 3D shapes. We downsample the number of frames for each scene by 5 to faciliate evaluation. We use a threshold $d$ of 0.05m for the F1 reconstruction metric calculation. For TSDF fusion, we set the parameters voxel size to 0.02m.

## C  EXPERIMENTAL SETUP

### C.1  TRAING DATASETS

We provide our training datasets in Table 9. Note that for datasets with potential overlap between training and testing (ScanNet++ and DL3DV), we ensure a strict separation at the scene level, i.e., scenes in training and testing are mutually exclus

Table 9: Datasets used in Depth Anything 3 , including number of scenes, data type, and usage.

| Usage | Dataset | #Scenes | Data Type |
|---|---|---|---|
| Pose-geometry benchmark | HiRoom (ours) | 29 | Synthetic |
| | ETH3D (Schops et al., 2017) | 11 | LiDAR |
| | DTU (Aanæs et al., 2016) | 22 | LiDAR |
| | 7Scenes (Shotton et al., 2013) | 7 | LiDAR |
| | ScanNet++ (Yeshwanth et al., 2023) | 20 | LiDAR |
| Pose-geometry Training | AriaDigitalTwin (Pan et al., 2023) | 237 | Synthetic |
| | AriaSyntheticENV (Pan et al., 2023) | 99950 | Synthetic |
| | ArkitScenes (Baruch et al., 2021) | 4388 | LiDAR |
| | BlendedMVS (Yao et al., 2020) | 503 | 3D Recon |
| | Co3dv2 (Reizenstein et al., 2021) | 30616 | Colmap |
| | DL3DV (Ling et al., 2024) | 6379 | Colmap |
| | HyperSim (Roberts et al., 2021) | 344 | Synthetic |
| | MapFree (Arnold et al., 2022) | 921 | Colmap |
| | MegaDepth (Li and Snavely, 2018) | 268 | Colmap |
| | MegaSynth (Jiang et al., 2025) | 6049 | Synthetic |
| | MvsSynth (Huang et al., 2018) | 121 | Synthetic |
| | Objaverse (Deitke et al., 2023) | 505557 | Synthetic |
| | Omniobject (Wu et al., 2023) | 5885 | Synthetic |
| | PointOdyssey (Zheng et al., 2023) | 44 | Synthetic |
| | ReplicaVMAP (Straub et al., 2019b) | 17 | Synthetic |
| | ScanNet++ (Yeshwanth et al., 2023) | 230 | LiDAR |
| | ScenenetRGBD (McCormac et al., 2017) | 16866 | Synthetic |
| | TartanAir (Wang et al., 2020) | 355 | Synthetic |
| | Trellis (Xiang et al., 2024) | 557408 | Synthetic |
| | vKitti2 (Cabon et al., 2020) | 50 | Synthetic |
| | WildRGBD (Xia et al., 2024) | 23050 | LiDAR |
| NVS Training | DL3DV (Ling et al., 2024) | 10015 | Colmap |
| NVS Benchmark | Tanks and Temples (Knapitsch et al., 2017a) | 6 | Colmap |
| | MegaDepth (Li and Snavely, 2018) | 19 | Colmap |
| | DL3DV (Ling et al., 2024) | 140 | Colmap |

### C.2  TRAINING DETAILS

We train our model on 128 H100 GPUs for 200k steps, using an 8k-step warm-up and a peak learning rate of $2 \times 10^{-4}$. Training image resolutions are randomly sampled from $504 \times 504$, $504 \times 378$,

Table 10: **Comparison of Models with Parameters and Running Speed.** The maximum number of images was tested on an 80 GB A100 GPU. If we store some intermediate tokens in CPU memory, we could process many more images. The running speed was measured on an A100 GPU with a scene of 32 images, and we report the average speed per image.

| Model | Max # of Images | Parameters | | | Running Speed |
|---|---|---|---|---|---|
| | | Backbone | DualDPT | CameraHead | |
| VGGT(Reference) | 400-500 | 0.91B | 0.064B | 0.22B | 34.1 FPS |
| DA3-Giant | 900-1000 | 1.130B | 0.050B | 0.48B | 37.6 FPS |
| DA3-Large | 1500-1600 | 0.300B | 0.047B | 0.21B | 78.37 FPS |
| DA3-Base | 2100-2200 | 0.086B | 0.045B | 0.12B | 126.5 FPS |
| DA3-Small | 4000-4100 | 0.022B | 0.043B | 0.03B | 160.5 FPS |

Table 11: **Ablation studies on teacher model geometry.** Depth-based geometry achieves $\delta_1$ comparable to disparity, while attaining the best AbsRel and SqRel among the three geometry representations.

| Geometry | $\delta_1$ ($\uparrow$) | AbsRel ($\downarrow$) | SqRel ($\downarrow$) |
|---|---|---|---|
| Disparity | **0.919** | 0.095 | 1.033 |
| Pointmap | 0.912 | 0.096 | 0.693 |
| Depth | 0.918 | **0.089** | **0.637** |

$504 \times 336$, $504 \times 280$, $336 \times 504$. For the $504 \times 504$ resolution, the number of views is sampled uniformly from [2, 18]. The batch size is dynamically adjusted to keep the token count per step approximately constant. Supervision transitions from ground-truth depth to teacher-model labels at 120k steps. Pose conditioning is randomly activated during training with probability 0.1.

### C.3 BASELINES

VGGT (Wang et al., 2025a) is an end-to-end transformer that jointly predicts camera parameters, depth, and 3D points from one or many views. Pi3 (Wang et al., 2025d) further adopts a permutation-equivariant design to recover affine-invariant cameras and scale-invariant point maps from unordered images. MapAnything (Keetha et al., 2025) provides a feed-forward framework that can also take camera pose as input for dense geometric prediction. Fast3R (Yang et al., 2025b) extends point-map regression to hundreds or even thousands of images in a single forward pass. Finally, DUSt3R (Wang et al., 2024c) tackles uncalibrated image pairs by regressing point maps and aligning them globally. Our method is similar to VGGT (Wang et al., 2025a), but adopts a new architecture and a different camera representation, and it is orthogonal to Pi3 (Wang et al., 2025d).

## D ADDITIONAL ANALYSIS

We present additonal analysis on Parameters, max number of images and running speed in Tab. 10

## E ADDITIONAL EXPERIMENTS

### E.1 TEACHER MODEL

**Teacher model training.** We ablate the teacher using a ViT-L backbone with batch size 64. Evaluation follows the DA2 benchmark protocol, and we additionally report Squared Relative Error (SqRel), defined as the mean squared error between predictions and ground truth normalized by the ground truth. Across geometries (Tab. 11), depth emerges as the most effective target compared with disparity and point maps. For training objectives (Tab. 12), the full teacher loss proposed in this work outperforms both the DA2 loss and a variant without proposed normal-loss term. Finally, data scaling contribute notably to performance (Tab. 13): upgrading datasets from V2 to V3 and adopting a multi-resolution training strategy yield consistent improvements in the teacher's final metrics.

Table 12: **Ablation studies on teacher model loss.** The full teacher-loss configuration yields the strongest performance, outperforming the other two loss variants across all metrics.

| Loss | $\delta_1$ ($\uparrow$) | AbsRel ($\downarrow$) | SqRel ($\downarrow$) |
|---|---|---|---|
| MAE-Loss | 0.918 | 0.089 | 0.637 |
| Teacher-Loss w/o Dist. Nor. | 0.918 | **0.087** | 0.600 |
| Full teacher-loss | **0.919** | **0.087** | **0.596** |

Table 13: **Ablation studies on teacher model data.** V2 denotes the datasets used to train the DA2 teacher model. V3 denotes those used for the DA3 teacher model. Traning with V3 datasets and multi-resolution strategy improves teacher model performance.

| Data | $\delta_1$ ($\uparrow$) | AbsRel ($\downarrow$) | SqRel ($\downarrow$) |
|---|---|---|---|
| V2 | 0.919 | 0.087 | 0.596 |
| V3 | 0.929 | 0.079 | 0.508 |
| V3 + multi-res. | **0.938** | **0.072** | **0.452** |

## E.2 ADDITIONAL ABLATIONS FOR DEPTH ANYTHING 3

**Dual-DPT Head.** We assess the effectiveness of the dual-DPT head via an ablation in which two separate DPT heads predict depth and ray maps independently. Results are reported in Tab. 14, item (d). Compared with the model equipped with the dual-DPT head, the variant without it shows consistent drops across metrics, confirming the effectiveness of our dual-DPT design.

**Teacher model supervision.** We ablate the use of teacher model labels as supervision, with quantitative results reported in Tab. 14, item (e). Training without teacher labels yields a slight improvement on DTU but leads to performance drops on 7Scenes and ScanNet++. Notably, the degradation is pronounced on HiRoom. We attribute this to HiRoom's synthetic nature and its ground truth containing abundant fine structures; supervision from the teacher helps the student capture such details more accurately. Qualitative comparisons in Fig. 8 corroborate this trend: models trained with teacher-label supervision produce depth maps with substantially richer detail and finer structures.

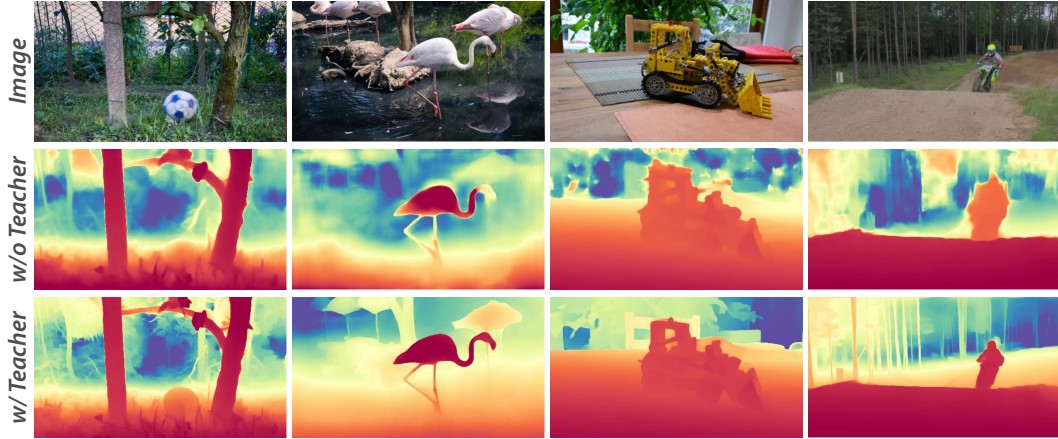

Figure 8: **Comparison of teacher-label supervision.** Supervision with teacher-generated labels yields depth maps with substantially richer detail and finer structures.

**Pose conditioning.** To assess the pose-conditioning module, we ablate it on the ViT-L backbone and report results in Tab. 14, items (f) and (g). Unlike other entries in the table, these two are evaluated with ground-truth pose fusion (marked with "*"), whereas the rest use predicted pose fusion. Across metrics, configurations with pose conditioning consistently outperform those without, confirming the effectiveness of the pose-conditioning module.

Table 14: **More ablations for single transformer.** We evaluate the effects of the dual-DPT head, teacher label supervision, and the pose conditioning module. Methods marked with "*" are evaluated with ground-truth pose fusion.

| Methods | HiRoom | | ETH3D | | DTU | | 7Scenes | | ScanNet++ | |
|---|---|---|---|---|---|---|---|---|---|---|
| | Auc3↑ | F1↑ | Auc3↑ | F1↑ | Auc3↑ | CD↓ | Auc3↑ | F1↑ | Auc3↑ | F1↑ |
| a. Full Model | 39.2 | 47.0 | 21.0 | 55.4 | 45.8 | 3.82 | 26.2 | 47.6 | 30.3 | 51.1 |
| d. w/o Dual DPT | 5.59 | 11.5 | 13.6 | 33.4 | 21.7 | 5.14 | 14.2 | 49.4 | 26.5 | 46.6 |
| e. w/o Teacher | 11.2 | 16.0 | 16.2 | 57.6 | 52.5 | 3.29 | 23.3 | 40.3 | 26.2 | 47.7 |
| f. w/o Pose Cond.* | | 65.8 | | 63.2 | | 3.65 | | 58.4 | | 62.8 |
| g. w/ Pose Cond.* | | 73.8 | | 70.9 | | 2.14 | | 46.0 | | 65.7 |

### E.3 ADDITIONAL COMPARISONS FOR VISUAL RENDERING

**Additional implementation details.** We retrain all compared feed-forward 3DGS models, ensuring that the training configuration matches the testing setup by using 12 input context views selected through farthest point sampling. We apply engineering optimizations such as flash attention and fully shared data parallelism to enable all models to process 12 input views efficiently. Depth training loss are incorporated for all baselines to ensure stable training and convergence. All models are trained on 8 A100 GPUs for 200K steps with a batch size of 1, except for pixelSplat, which is trained for 100K steps due to rather slow epipolar attention. All results are reported at $H \times W = 270 \times 480$.

**Visual quality analysis.** We present visual comparisons with other models in Fig. 9 under novel view synthesis settings. As illustrated, simply augmenting our DA3 model with a 3D Gaussian DPT head yields significantly improved rendering quality over existing state-of-the-art approaches. Our model demonstrates particular strength in challenging regions, such as thin structures (*e.g.*, columns in the first and third scenes) and large-scale outdoor environments with wide-baseline input views (last two scenes), as shown in Fig. 9. These results underscore the importance of a robust geometry backbone for high-quality visual rendering, consistent with our quantitative findings in Tab. 4. We anticipate that the strong geometric understanding of DA3 will also benefit other 3D vision tasks.

### DISCLOSURE

The authors acknowledge the use of large language models (LLMs) solely for grammar checking and language refinement of this manuscript. The input images in the teaser demo were extracted from a publicly available YouTube video (Drones, 2024), credited to the original creator.

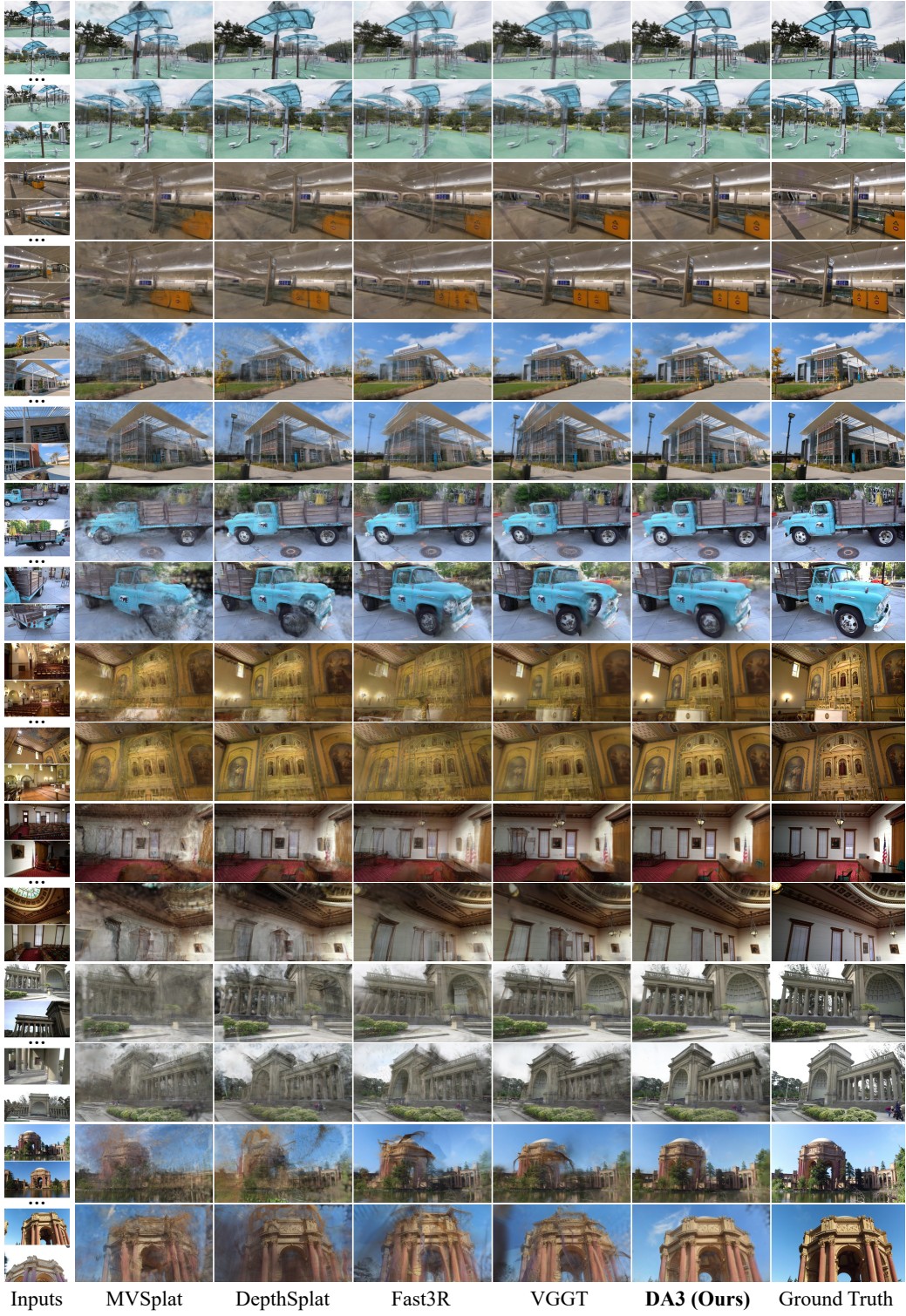

| Inputs | MVSplat | DepthSplat | Fast3R | VGGT | **DA3 (Ours)** | Ground Truth |

Figure 9: **Qualitative comparisons with state-of-the-art methods for visual rendering.** The first column shows the selected input views, while the remaining columns display novel views rendered by comparison models and ground truth. For each scene, two rendered novel viewpoints are presented in consecutive rows. The first three scenes are from DL3DV, the following two are from Tanks and Temples, and the last three are from MegaDepth. Compared to other methods, our model consistently achieves superior rendering quality across diverse and challenging scenes.

