# OpenReview forum: "Depth Anything 3: Recovering the Visual Space from Any Views"
_ICLR.cc/2026/Conference — ICLR 2026 Oral_

### Official Review · Reviewer_Yy63 · 2025-11-01

**Soundness:** 3
**Presentation:** 3
**Contribution:** 3
**Rating:** 6
**Confidence:** 3

**Summary:**

The paper presents Depth Anything 3 (DA3), which learns to predict dense depth maps and ray maps for estimating 3D geometry and camera poses. DA3 focuses in simplifying geometric understanding both from its model architecture, and also by using depth ray as representation for prediction. The architecture simply adopts a pre-trained DINOv2, which has been powerful in various 2D and 3D tasks, and rearranges the tokens for certain layers to compute the full attention for image tokens across different frames, allowing the information exchange between views. Furthermore, DA3 predicts camera rays instead of point maps, which consists of the camera origin and the direction for each of the pixels, and thoroughly demonstrates that camera rays serve as better representations compared to point clouds.

**Strengths:**

1. The paper shares an interesting finding that understanding 3D geometry can be done in a simplistic manner, especially without specialized architecture for incorporating multiple views or enforcing geometric constraints. Despite that it has been a recent trend for applying more and more generic architecture in 3D tasks, it is very intriguing to see that it could be done within a pre-trained DINOv2 by tweaking some of its attention layers.

2. The paper formulates a novel framework for predicting camera rays, which is the ray connecting the camera origin and pixels in the image plane. The results and the ablations thoroughly show that the combination of depth and camera ray suffices for effectively understanding 3D geometry,

3. The proposed method establishes state-of-the-art performance across various geometry tasks, while having similar parameter counts. Furthermore, the method also shows strength in efficiency thanks to its simplistic architecture with having only few layers for cross-view understanding.

4. The paper proposes a benchmark for evaluating visual geometry, and introduces HiRoom dataset which the authors will release in the future.

**Weaknesses:**

1. Some of the claims are not well justified, or seems to be overclaiming in some points.
- L.157-158 (While point maps are insufficient to ensure consistency, redundant targets can improve pose accuracy but often introduce entanglement that compromises it.) What does the authors intend with "entanglement"? Does this mean that the results deduced from different heads could be problematic (e.g. pose from pose prediction head v. pose from point maps)? The authors should better elaborate the problems of "redundant" prediction to better establish their motivation for predicting camera rays.
- The camera head, despite being optional, is specified to have 0.48B parameters for the Giant variant. Despite the authors show that the computation is negligible as it only has few tokens, having a camera head that has nearly half of the parameter count of the backbone does not seem "lightweight".

2. It is a bit unclear on why depth+ray is more effective compared to point maps. To the reviewer's understanding, depth and ray actually seems like decoupling the point map prediction from Dust3r into two separate predictions. Why would this be better, asides from the empirical results?

3. Although the teacher model seems crucial, the ablations seems to be missing.

**Questions:**

1. Considering that all pixels from the same camera should have identical origins, it is interesting to see that all of the pixels are required to predict the origin in a dense manner, in addition to the ray direction. Is this simply a design choice, or does this have impact on training? It would also be interesting to see the variance of the predicted origin across each pixels within a single image, and study whether averaging all of the pixels for the pose estimation strategy is helpful.

2. The idea for the architecture seems to share some ideas with ViTDet[1], as it modifies existing pre-trained ViTs to alternate between local/global attentions. However, it is interesting to see that limiting the blocks for cross-view attention was more effective for DA3, as opposed to the results from ViTDet, apart from the obvious efficiency gains. Could the authors provide further analysis on why Full Alt. performs worse? Could it be from the gap between the pre-training and the downstream task, where drastically modifying all of the layers within DINOv2 to all handle cross-view attention be problematic?

[1] Li, Yanghao, et al. "Exploring plain vision transformer backbones for object detection." European conference on computer vision. Cham: Springer Nature Switzerland, 2022.

---

> ### Author Response · Authors · 2025-11-20
>
> We thank the reviewer for the constructive feedback. We have clarified the motivation behind our design choices and revised the relevant descriptions in the paper as detailed below.
>
> **1. Entanglement and redundancy.** We acknowledge that our original description was not sufficiently clear. A point map is an entangled representation of depth and rays (camera pose), and prior works [VGGT, Fast3R] have shown that it is not more effective than depth alone for representing geometry. Therefore, additional depth heads are introduced [Fast3R, VGGT, Cut3R], which creates redundancy for pointmap. We have clarified our description in the revised paper.
>
> **2. Lightweight camera head.** We acknowledge that the camera head is not lightweight in terms of parameter count, and we have revised our description accordingly. The head directly follows the design of VGGT, and our original use of “lightweight” referred specifically to its small computational cost (FLOPs), as it operates on very few tokens. Moreover, we have clarified that the camera head is entirely optional and is only used when faster camera inference is desired; the core representation of DA3 remains depth + rays.
> **3. Why ray + depth.**
> - We would like to clarify that our primary motivation (Sec. 2.1) is to identify a minimal set of prediction targets for effective modeling. This is mainly achieved through empirical study, with the experimental results (Table 4) showing that depth + ray gives the best performance among four combinations.
> - We hypothesize that our depth-ray representation provides the following benefits:
>     - **Decoupling of Geometry and Camera:** By predicting depth and rays separately, the network can learn to model scene structure and camera motion independently. This decomposition might simplify the overall learning task compared to end-to-end approaches that must solve for both simultaneously.
>     - **An Effective Intermediate Pose Representation:** Ray prediction strikes a balance between expressiveness and efficiency. It is a dense, fine-grained representation of the camera pose that provides richer supervision than a single vector, yet it avoids the informational redundancy of a full
>     point-map prediction, which conflates pose with geometry.
>
> **4. Influence of the teacher.** We present both quantitative and qualitative results in the supplementary material (Table 12 and Figure 8). The teacher model improves quality in both aspects, especially on HiRoom (a high-quality synthetic dataset), and yields slight improvements on the other datasets.
>
> **5. Pixel-wise ray map origin.** We experimentally observed that the predicted ray origins exhibit extremely small variance and are essentially constant across pixels. We have added an additional experiment where the ray origin is predicted with a single MLP; this generally leads to a slight performance drop (e.g., 35.12 vs. 32.22 in terms of AUC3), supporting the robustness benefit of the dense formulation.
>
> **Ablation study on ray origin prediction.**
>
> | Method                | Avg  | HiRoom | ETH3D | DTU  | 7Scenes | ScanNet++ |
> |-----------------------|------|--------|-------|------|---------|-----------|
> | depth + ray + cam     | 35.1 | 37.2   | 22.3  | 56.3 | 25.7    | 34.1      |
> | depth + ray-st + cam  | 32.2 | 28.7   | 22.7  | 48.8 | 27.0    | 33.9      |
>
> **6. Why full alt. performs worse.** A plausible explanation is that the early transformer blocks function primarily as feature extractors, and preserving their intra-view structure benefits the later cross-view interaction. Full alternation introduces cross-view mixing from the very first layers, which interferes with low-level feature extraction and results in weaker performance.

---

> > ### Author Response · Authors · 2025-11-28
> >
> > Dear Reviewer Yy63,
> >
> > Thank you again for your thoughtful and constructive review. We have carefully addressed the points you raised and added additional clarifications and experimental results in our official response.
> >
> > If there is any remaining confusion or if further details would be helpful, we would be more than happy to provide additional clarification. We sincerely appreciate your time and feedback, and we look forward to hearing any further comments you may have.
> >
> > Thank you for your consideration.

---

> ### Comment · Reviewer_Yy63 · 2025-11-28
>
> I have read through the rebuttals from the authors, as well as the discussions with the other reviewers.
>
>
> Overall, I find the response from the authors to provide clear, sufficient answers to my initial concerns. The benefits of the depth-ray configuration suggested by the authors are very convincing, and the revised manuscript provides a much clearer understanding on how the paper is tackling redundancy. I also agree that the camera head the camera head can be considered lightweight in terms of FLOPS, and understand that it follows the camera head from VGGT. Although there are some variance across different datasets, the additional experiment provided by the authors seems enough to demonstrate how the dense formulation is beneficial during training.
>
>
> With all of these in consideration, I will raise my rating to 8. Unfortunately, it seems like openreview is not allowing the reviewers to adjust the rating at the moment, so I will reflect the rating once it becomes possible.

---

### Official Review · Reviewer_4H8r · 2025-11-01

**Soundness:** 3
**Presentation:** 3
**Contribution:** 3
**Rating:** 6
**Confidence:** 4

**Summary:**

The authors address the problem of 3D geometry estimation. They argue that depth and ray predictions constitute the minimal set of 3D predictions necessary for rendering 3D geometry, demonstrating this is the optimal choice. They extend existing transformer architectures by introducing a cross-view interaction transformer layer to handle multi-view inputs. Their method achieves significantly superior performance compared to existing models in both pose estimation and geometry estimation.

**Strengths:**

- The paper utilized depth and ray map representations to enable full 3D reconstruction from an arbitrary number of input images.
- Discovered an effective architecture design that outperforms previous methods while requiring minimal modifications to DINOv2.
- The paper demonstrates their model's effectiveness across various experimental settings.

**Weaknesses:**

- **Unclear advantage of depth+ray over point map:** To my knowledge, point maps can effectively represent various 3D information such as depth and pose, and a point map is essentially a combination of depth and ray maps. However, Table 5 shows that point maps hurt pose accuracy. What is the reason for this performance degradation? This finding appears to contradict the ablation study in VGGT, which argues that point map accuracy increases with multimodal outputs. I would like to see a more comprehensive analysis explaining why the combination of ray and depth maps outperforms point maps.
- **Missing ablation studies with point maps:** I am curious about additional experiments in Table 5 calculate the metrics using point map representations not using ray map and depth map. Specifically, what are the results when training with: (1) point maps only, and (2) point maps combined with ray maps and depth maps?
- **Pixel-wise ray map origin justification:** I understand that the origin of the ray map is identical for each image. Is there a specific reason to set the ray map origin in a pixel-wise manner? What is the benefit of this design choice?
- **Distinction between camera head and ray predictions:** What is the precise difference between the camera head and ray predictions? I am also curious about the performance gap between these two approaches. To my knowledge, camera parameters can generate ray maps, and conversely, ray maps can also be used to estimate camera parameters. Could you clarify the relationship and trade-offs between these representations?

**Questions:**

Please see the weakness section.

---

> ### Author Response · Authors · 2025-11-20
>
> We thank the reviewer for the insightful comments. We have conducted additional ablation studies and provided detailed explanations regarding the design choices of our depth-ray representation below.
>
> **1. Performance degradation on PointMap.** We would like to clarify that there is no average performance degradation for PointMap. Table 4 shows that depth + pcd + cam consistently outperforms depth + cam on ETH3D, DTU, 7Scenes, and ScanNet++, with only a slight drop on HiRoom.
>
> **2. Why ray + depth.**
> - We would like to clarify that our primary motivation (Sec. 2.1) is to identify a minimal set of prediction targets for effective modeling. This is mainly achieved through empirical study, with the experimental results (Table 4) showing that depth + ray gives the best performance among four combinations.
> - We hypothesize that our depth-ray representation provides the following benefits:
>     - **Decoupling of Geometry and Camera:** By predicting depth and rays separately, the network can learn to model scene structure and camera motion independently. This decomposition might simplify the overall learning task compared to end-to-end approaches that must solve for both simultaneously.
>     - **An Effective Intermediate Pose Representation:** Ray prediction strikes a balance between expressiveness and efficiency. It is a dense, fine-grained representation of the camera pose that provides richer supervision than a single vector, yet it avoids the informational redundancy of a full point-map prediction, which conflates pose with geometry.
>
> **3. Additional ablation on pointmap.** Following the reviewer's suggestion, we added two additional baseline variants: **pcd** and **depth + ray + pcd**. Experiments demonstrate that pointmap performs worse than depth + ray. Furthermore, incorporating pointmap (depth + ray + pcd) does not significantly improve the performance of depth + ray.
>
> **Ablation study on pointmap variants in terms of pose accuracy (Auc3) and geometry accuracy (F-score/chamfer distance).**
>
> | Setting             | Avg | HiRoom | ETH3D | DTU  | 7Scenes | ScanNet++ |
> |---------------------|----------|--------|-------|------|---------|-----------|
> | pcd                 | 31.6    | 35.9   | 20.3  | 49.1 | 21.7    | 30.8      |
> | depth + ray + pcd   | 36.4    | 52.3   | 22.6  | 43.0 | 27.3    | 36.7      |
> | **depth + ray**     | 36.0    | 48.7   | 25.5  | 46.5 | 24.0    | 35.5      |
>
> | Setting             | Avg | HiRoom | ETH3D | 7Scenes | ScanNet++ | DTU CD |
> |---------------------|--------|--------|-------|---------|-----------|--------|
> | pcd                 | 51.5  | 42.7   | 64.6  | 45.5    | 53.3      | 4.219  |
> | depth + ray + pcd   | 56.5  | 61.4   | 64.2  | 48.5    | 51.8      | 4.158  |
> | **depth + ray**     | 56.4  | 60.3   | 65.4  | 46.5    | 53.4      | 3.919  |
>
>
> **4. Pixel-wise ray map origin.** We experimentally observed that the predicted ray origins exhibit extremely small variance and are essentially constant across pixels. We have added an additional experiment where the ray origin is predicted with an MLP; this generally leads to a slight performance drop (e.g., 35.12 vs. 32.22 in terms of AUC3), supporting the robustness benefit of the dense formulation.
>
> **Ablation study on ray origin prediction.**
>
> | Method                | Avg  | HiRoom | ETH3D | DTU  | 7Scenes | ScanNet++ |
> |-----------------------|------|--------|-------|------|---------|-----------|
> | depth + ray + cam     | 35.1 | 37.2   | 22.3  | 56.3 | 25.7    | 34.1      |
> | depth + ray-st + cam  | 32.2 | 28.7   | 22.7  | 48.8 | 27.0    | 33.9      |
>
>
>
> **5. Camera head and ray predictions.** Ray prediction is necessary for achieving high pose accuracy during training process, while the camera head is optional and retained only for faster inference. Specifically, direct camera head prediction takes 0.46 ms, whereas solving pose from the ray map requires approximately 8.60 ms (measured on an A100 GPU, averaged over a 49-image scene; for reference, the feature forward pass with image token attention takes 43 ms).
>
> **Comparison between camera head and ray prediction for DA3-Giant.**
>
> | Method     | Time (ms) | Avg | HiRoom | ETH3D | DTU  | 7Scenes | ScanNet++ |
> |------------|-----------|---------|--------|-------|------|---------|-----------|
> | Camera     | 0.46      | 63.8    | 81.7   | 39.3  | 85.6 | 29.2    | 83.2      |
> | Ray        | 8.60      | 68.0    | 88.6   | 42.4  | 89.3 | 29.6    | 89.9      |

---

> > ### Comment · Reviewer_4H8r · 2025-11-27
> >
> > I appreciate the extensive experiments and detailed responses. All my concerns have been addressed, so I am raising my score.

---

> > > ### Author Response · Authors · 2025-11-28
> > >
> > > We are grateful that our responses have fully addressed your concerns. Please feel free to let us know if there is anything else we can further clarify.

---

### Official Review · Reviewer_vvhC · 2025-11-01

**Soundness:** 4
**Presentation:** 4
**Contribution:** 3
**Rating:** 8
**Confidence:** 4

**Summary:**

This paper introduces Depth Anything 3. Different from Depth Anything and Depth Anything v2 that can only work with single image, Depth Anything 3 is able to process any number of images. Different from previous methods such as DUSt3R and VGGT, Depth Anything 3 simplify the architecture, making it more scalable to numerous images. In addition, a teacher-student paradigm is used to provide high-quality data. Depth Anything 3 achieves state-of-the-art performance in various tasks, including pose estimation, 3D reconstruction, and rendering.

**Strengths:**

1. The architecture of Depth Anything 3 is simpler than previous methods. Depth Anything 3 uses a single vision transformer, while previous methods typically use vision transformer and following self- & cross-attention. Input-adaptive self-attention is used in vision transformer to enable cross-view attention without introducing new attention layers. With a simpler structure, Depth Anything 3 is able to process more images, which is meaningful for the future research.

2. Extensive and thorough evaluation. Performance of pose estimation, 3D reconstruction, and rendering are thoroughly evaluated, where Depth Anything 3 achieves state-of-the-art performance.

3. Ablation study of depth-ray representation shows that it explicitly outperforms previous representations, e.g. depth+pcd+cam used by VGGT.

**Weaknesses:**

1. In Table 1 and Table 2, I recommend adding some state-of-the-art methods that are not feed-forward models. This can help the readers have a better understanding of the performance difference between different methods. For example, classical pipelines generally outperform feed-forward models in 3D reconstruction.

2. If the teacher is not used, would the performance degrade explicitly? Currently, I am not sure if the mainly improvement is from the powerful teacher.

**Questions:**

1. L142: the equation looks wrong. $P$ denotes the 3D point in world coordinate frame, $D_i(u,v) K_i^{-1} p$ denotes the 3D point in camera local frame. To make the equation correct, $R_i, t_i$ should represent camera pose (transformation from camera to world), instead of extrinsics (world to camera).

2. Sec. 2.4: Is GS-DPT head the only optimizable module, i.e. backbone is fixed?

3. L967: On ETH3D, is the tolerance 0.25 meter? Could the authors provide individual performance on each scene since the scale of scene vary a lot?

4. Typo:
    * L823: a “identity” camera -> an “identity” camera

---

> ### Author Response · Authors · 2025-11-20
>
> We thank the reviewer for the thorough evaluation and valuable suggestions. We have added classical baselines and clarified several technical details as requested below.
>
> **1. Adding classical baselines.** We agree that classical methods help contextualize performance, and we have added representative COLMAP [Schonberger et al., CVPR 2016] and GLOMAP [Pan et al., ECCV 2024] pipelines to Tables 1–2 so that readers can directly compare DA3 to non-feed-forward methods. COLMAP performs very well on DTU, which is extremely dense (1.67 mm vs. 1.92 mm CD error, COLMAP vs. Ours), but its performance degrades significantly on datasets that are even slightly sparser. For example, on ETH3D, COLMAP + PatchMatchStereo achieves an F-score of 20.7 compared to 74.4 from our method.
>
>
> **2. Influence of the teacher.** We present both quantitative and qualitative results in the supplementary material (Table 12 and Figure 8). The teacher model improves quality in both aspects, especially on HiRoom (a high-quality synthetic dataset), and yields slight improvements on the other datasets.
>
> **3. Notations.** **R** and **t** denote a camera-to-world pose. We have made this clearer in the revision.
>
> **4. GS-DPT.** Yes, only GS-DPT is trainable. The backbone is initialized with the DA3 geometry model and kept frozen during the FF3DGS fine-tuning process.
>
> **5. ETH3D tolerance and per-scene results.** The tolerance of all scenes is 0.25 m. We have included a per-scene ETH3D breakdown as in Table 3.
>
>
>
> **Table 1. Pose accuracy in terms of AUC3.**
>
> | Methods      | Pico  | ETH3D | DTU  | 7Scenes | ScanNet++ |
> |--------------|-------|-------|------|---------|-----------|
> | COLMAP       | 12.98 | 4.75  | 87.2 | 22.3 | 13.3 |
> | GLOMAP       | 31.9  | 8.37  | 96.8 | 24.1 | 20.8 |
> | DUSt3R       | 17.6  | 4.30  | 4.00 | 6.90 | 8.10 |
> | Fast3R       | 25.9  | 8.10  | 9.50 | 19.0 | 17.9 |
> | MapAnything  | 17.9  | 19.2  | 6.50 | 12.6 | 20.2 |
> | Pi3          | 67.0  | 35.2  | 62.5 | 25.5 | 50.7 |
> | VGGT         | 49.1  | 26.3  | 79.2 | 23.9 | 62.6 |
> | DA3-Giant    | 81.7  | 39.3  | 85.6 | 29.2 | 83.2 |
> | DA3-Large    | 37.9  | 19.0  | 58.4 | 25.1 | 46.9 |
> | DA3-Base     | 12.8  | 13.6  | 31.4 | 17.2 | 16.2 |
> | DA3-Small    | 3.40  | 4.89  | 9.46 | 6.19 | 2.86 |
>
>
>
> **Table 2. Geometry accuracy is reported in terms of F-score, except for DTU, where we use CD (mm).**
>
> | Methods       | Pico | ETH3D | DTU  | 7Scenes | ScanNet++ |
> |---------------|------|-------|------|---------|-----------|
> | COLMAP+PM     | 16.8 | 20.7 | 1.67 | 40.0 | 15.7 |
> | GLOMAP+PM     | 30.7 | 24.1 | 1.62 | 43.9 | 16.5 |
> | DUSt3R        | 30.1 | 19.7 | 7.60 | 26.6 | 18.9 |
> | Fast3R        | 40.7 | 38.5 | 6.88 | 41.0 | 37.1 |
> | MapAnything   | 32.4 | 54.8 | 7.91 | 44.8 | 39.4 |
> | Pi3           | 75.8 | 72.7 | 3.28 | 44.2 | 63.1 |
> | VGGT          | 56.7 | 57.2 | 2.05 | 47.9 | 66.4 |
> | DA3-Giant     | 89.3 | 74.4 | 1.92 | 52.0 | 76.4 |
> | DA3-Large     | 48.2 | 57.3 | 3.45 | 48.7 | 58.9 |
> | DA3-Base      | 18.6 | 52.8 | 5.14 | 37.8 | 39.7 |
> | DA3-Small     | 12.9 | 39.4 | 5.12 | 30.8 | 24.2 |
>
>
>
> **Table 3. Per-scene breakdown on ETH3D dataset in terms of F-score.**
>
> | Scene         | DA3-Giant | DA3-Large | VGGT |
> |---------------|-----------|-----------|------|
> | courtyard     | 91.0      | 14.1      | 37.7 |
> | electro       | 75.7      | 54.6      | 64.9 |
> | kicker        | 99.9      | 99.3      | 99.8 |
> | pipes         | 81.7      | 64.9      | 83.5 |
> | relief        | 64.6      | 31.1      | 19.1 |
> | delivery_area | 11.8      | 41.6      | 25.5 |
> | facade        | 71.4      | 44.5      | 59.8 |
> | office        | 99.4      | 95.6      | 99.2 |
> | playground    | 76.8      | 68.1      | 73.1 |
> | relief_2      | 53.2      | 46.3      | 1.19 |
> | terrains      | 92.4      | 68.4      | 66.6 |
> | **Mean**      | 74.4      | 57.3      | 57.2 |

---

> > ### Author Response · Authors · 2025-11-28
> >
> > Dear Reviewer vvhC, we hope our responses have addressed your questions. Is there anything that remains unclear or that you would like us to elaborate on further?

---

### Official Review · Reviewer_xgar · 2025-11-02

**Soundness:** 3
**Presentation:** 3
**Contribution:** 3
**Rating:** 8
**Confidence:** 3

**Summary:**

The paper presents Depth Anything 3, a single model that unifies geometric understanding across any number of views. The method jointly predicts a depth map and a ray map, using a ViT backbone with input adaptive cross view attention. A Dual DPT head shares reassembly modules and branches only at the final fusion stage to jointly infer depth and rays. Experiments across diverse benchmarks show consistent state of the art results in pose estimation, geometric reconstruction, and feed forward novel view synthesis, demonstrating strong accuracy and efficiency.

**Strengths:**

1. This paper presents a thoughtful analysis of what modalities are truly necessary for strong vision understanding tasks. It argues that depth together with a ray map is a minimal and sufficient target set. The ablation in Table 5 convincingly supports this claim by outperforming alternatives. Although recent work MapAnything also discusses incorporating ray maps into a unified representation, it is a contemporaneous work and does not needed to be considered here.

2. The Dual DPT head is well designed. By sharing reassembly modules and branching only at the final fusion stage, the approach enforces pixel level alignment while avoiding redundant representations, which benefits both accuracy and efficiency.

3. The experimental study is extensive and persuasive. The method is validated across pose estimation, geometric reconstruction, and feed forward novel view synthesis, consistently achieving SOTA results.

**Weaknesses:**

I did not find any major weaknesses. While I know recent advances in this area, I am not fully confident about all technical nuances and distinctions among closely related methods. I am open to perspectives from other reviewers and will continue to track the discussion.

**Questions:**

I wonder whether Depth Anything 3 is the most suitable title. The previous work is called Depth Anything v2, so if the intention is to follow the series it would be better to use Depth Anything v3 for consistency. Additionally, both v1 and v2 focus primarily on depth prediction. The current title can easily be read as another improvement targeted at depth prediction. It may be worth considering an alternative title that more clearly conveys the contribution..

---

> ### Author Response · Authors · 2025-11-20
>
> We thank the reviewer for the positive feedback and constructive suggestions. We address the naming convention concern below.
>
> **1. Title: DA3 vs. DAv3.** We use Depth Anything 3 (DA3) because this work expands from monocular depth prediction to a unified multi-view geometric capability. The paper "Depth Anything v2" only focuses on monocular depth, while our work represents a broader conceptual extension.

---

> > ### Author Response · Authors · 2025-11-28
> >
> > Dear Reviewer xgar, we hope our responses have clarified your earlier questions. Please let us know if anything remains unclear or if you would like further details. Thank you for your consideration.

---

### Public Comment · ~Dany_Li1 · 2025-11-18
**Comparison with AnySplat**

The proposed method supports pose-free 3DGS recovery, while comparison with these methods is not shown. Does the proposed method outperform methods like AnySplat and NoPoSplat? I believe this is an interesting yet critical question to evaluate its effectiveness.

---

> ### Author Response · Authors · 2025-11-20
>
> Thank you for your feedback regarding pose-free 3DGS comparisons.
>
> Our feed-forward Novel View Synthesis (NVS) benchmark, as stated in L259-L260, uses COLMAP-estimated ground-truth poses for all models, making it pose-conditioned. This setup ensures a more fair and well-established comparison to systematically evaluate how different geometry foundation models benefit the downstream NVS task.
> The key issue of the “pose-free” benchmark is that it is not well established how to fairly compare novel views in a “pose-free” setup. For example, NoPoSplat relies on injected intrinsics (requiring modifications to the foundation models) and post-optimization, while AnySplat resorts to post-alignment of predicted poses, yet not all 3D foundation models predict multi-view camera poses directly or with comparable accuracy. These factors can introduce unfairness, making pose-free evaluation unsuitable for our benchmarking, which primarily aims to serve as a downstream application to evaluate different foundation models.
>
> Hence, we leave the issue of benchmarking pose-free novel view settings for potential future work that more specifically targets pose-free feed-forward 3DGS, rather than 3D foundation models like our work.

---

> > ### Public Comment · ~Dany_Li1 · 2025-11-24
> >
> > Thanks for your detailed response, which has solved my confusion. I believe this work is a masterpiece in both novelty and performance.

---

### Meta-Review · Area_Chair_MQ1Z · 2026-01-07

**Summary:**

The reviewers’ main concerns were about whether the proposed representation was truly necessary and non-redundant compared to point maps, how much the performance depended on the teacher model, and whether the added complexity was justified by the gains. They also questioned the clarity of the design choices and the computational and practical implications of the approach.

**Reviewer Concerns:**

The rebuttal addressed most of the technical concerns, including clarifying the representation choice (depth + ray vs. point maps), the role of the camera head, the influence of the teacher model, and the architectural design decisions through additional analysis and ablations.

**Reviewer Scores:**

Two reviewers initially gave a score of 8. Of the remaining two reviewers, one(Reviewer 4H8r) explicitly stated that they would likely increase their score after the rebuttal, and the other (Reviewer Yy63) indicated that they would raise their score to 8. Therefore, the overall scores would likely increase slightly, with at least three reviewers at or near 8 after discussion.

---

### Decision · Program_Chairs · 2026-01-26

Accept (Oral)